


# A regional spatio-temporal analysis of large magnitude
# snow avalanches using tree rings
Erich Peitzsch[1,2*], Jordy Hendrikx[2], Daniel Stahle[1], Gregory Pederson[1], Karl Birkeland[3,2],
and Daniel Fagre[1]
[1] U.S. Geological Survey Northern Rocky Mountain Science Center, West Glacier, Montana, USA
[2] Snow and Avalanche Lab, Department of Earth Sciences, Montana State University, Bozeman, Montana,
USA
[3] U.S.D.A. Forest Service National Avalanche Center, Bozeman, Montana, USA
*epeitzsch@usgs.gov, 215 Mather Dr., West Glacier, MT, USA, 59936
**Abstract.** Snow avalanches affect transportation corridors and settlements worldwide. In many mountainous
regions, robust records of avalanche frequency and magnitude are sparse or non-existent. However,
dendrochronological methods can be used to fill this gap and infer historic avalanche patterns. In this study,
we developed a tree-ring based avalanche chronology for large magnitude avalanche events using
dendrochronological techniques for a portion of the northern United States Rocky Mountains. We used a
strategic sampling design to examine avalanche activity through time and across nested spatial scales (i.e.
from individual paths, four distinct sub-regions, and the region). We analysed 673 total samples from 647
suitable trees collected from 12 avalanche paths, from which 2,134 growth disturbances were identified over
years 1636 to 2017 Common Era (C.E.). Using existing indexing approaches, we developed a regional
avalanche activity index to discriminate avalanche events from noise in the tree-ring record. Large magnitude
avalanches common across the region occurred in 30 individual years and exhibited a median return interval
of approximately three years (mean = 5.21 years). The median large magnitude avalanche return interval (3-
8 years) and the total number of avalanche years (12-18) vary throughout the four sub-regions, suggesting
the important influence of local terrain and weather factors. We tested subsampling routines for regional
representation, finding that sampling eight random paths out of a total of 12 avalanche paths in the region
captures up to 83% of the regional chronology, whereas four paths capture only 43% to 73%. The greatest
value probability of detection for any given path in our dataset is 40% suggesting that sampling a single path
would capture no more than 40% of the regional avalanche activity. Results emphasize the importance of
sample size, scale, and spatial extent when attempting to derive a regional large magnitude avalanche event
chronology from tree-ring records.



## 1 Introduction

### 1.1 Background

Snow avalanches are hazardous to human safety and infrastructure (Mock et al., 2016; Schweizer, 2003) as well as an important landscape disturbance affecting mountain ecosystems (Bebi et al., 2009). In the United States an average of 27 people die in avalanche accidents each winter (CAIC, 2020). Avalanches, especially large magnitude events, also affect transportation corridors and settlements throughout the world. For example, avalanches impact numerous roadways and railroad corridors in the western United States (Armstrong, 1981; Hendrikx et al., 2014; Reardon et al., 2008). Consequently, understanding general avalanche processes and associated large magnitude avalanche return intervals is critical for local and regional avalanche forecasters, transportation agencies, and land use planners.

Long-term, reliable, and consistent avalanche observation records are necessary for calculating avalanche return intervals which can be used in infrastructure planning and avalanche forecasting operations. However, such records are often sparse or non-existent in many mountainous regions, including areas with existing transportation corridors. Thus, inferring avalanche frequency requires the use of dendrochronological methods to document damaging events or geomorphic response within individual trees at individual path to regional scales. Even in regions with historical records, tree-ring dating methods can be used to extend or validate uncertain historical avalanche records, which has led to the broad implementation of these methods in mountainous regions throughout the world (e.g. Corona et al., 2012; Favillier et al., 2018; Schläppy et al., 2014) .

Numerous studies reconstructed avalanche chronologies in the United States using tree-ring methods (Burrows and Burrows, 1976; Butler et al., 1987; Carrara, 1979; Hebertson and Jenkins, 2003; Potter, 1969; Rayback, 1998;). Butler and Sawyer (2008) provided a review of current methodologies and types of tree-ring responses used in avalanche dendrochronological studies. Favillier et al. (2018) provided a more recent comprehensive graphical summary of dendrochronological avalanche studies throughout the world. Numerous studies used dendrochronological techniques to develop avalanche chronologies for remote regions without historical avalanche records or areas with inconsistent avalanche observations (Butler and Malanson, 1985a; Germain et al., 2009; Reardon et al., 2008; Šilhán and Tichavský, 2017; Voiculescu et al., 2016), and many studies used these techniques to examine avalanches across space and time (Table 1).



**Table 1: List of previous avalanche-dendrochronological work *with more than one avalanche path* in study – to**
**place our regional work in context with other regional/multiple path studies. Number of samples, paths, growth**
**disturbances (GD), and spatial extent (linear distance between most distant avalanche paths in study area) are**
**included. For spatial extent, *NA* is reported in studies where spatial extent is not reported or could not be inferred**
**from maps in the published work. Where spatial extent is not reported directly in previous work, it is estimated**
**by using maps from the published work and satellite imagery.**

| Authors | Location | # Trees | # Samples | # Paths | Spatial Extent | # GD |
|---|---|---|---|---|---|---|
| Gratton et al. (2019) | Northern Gaspé Peninsula, Québec, Canada | 82 | 177 cores 65 x- sec | 5 | ~20 km | Not provided |
| Meseșan et al. (2018) | Parâng Mountains, Carpathians, Romania | 232 | 430 cores 39 x-sec 4 wedges | 3 | ~16 km | Not provided |
| Favillier et al. (2018) | Zermatt valley, Switzerland | 307 | 620 cores 60 x-sec | 3 | ~1 km | 2570 |
| Ballesteros-Canovas (2018) | Kullu district, Himachal Pradesh, India | 114 | Not Provided | 1 slope (multiple paths) | ~ 1 km | 521 |
| Pop et al.(2018) | Piatra Craiului Mountains, Romania | 235 | 402 cores 34 x-sec | 2 | ~ 2 km | 789 |
| Martin and Germain (2016) | White Mountains, New Hampshire | 450 | 350 cores 456 x-sec | 7 | ~10 km | 2251 |
| Voiculescu et al. (2016) | Făgăras massif, Carpathians, Romania | 293 | 586 cores | 4 | *NA* | 853 |
| Schläppy et al. (2015) | French Alps, France | 967 | 1643 cores 333 x-sec | 5 | ~100 km | 3111 |
| Schläppy et al. (2014) | French Alps, France | 297 | 375 cores 63 x-sec | 2 | ~100 km | 713 |
| Schläppy et al. (2013) | French Alps, France | 587 | 1169 cores 122 x-sec | 3 | ~100 km | 1742 |
| Casteller et al. (2011) | Santa Cruz, Argentina | 95 | ~95 x-sec | 9 | ~2 km | Not provided |
| Köse et al. (2010) | Katsomonu, Turkey | 61 | Not provided | 2 | ~ 500 m | Not provided |





| Muntán et al. (2009) | Pyrenees, Catalonia | NA | 448 | 6 | ~150 km | Not provided |
|---|---|---|---|---|---|---|
| Germain et al. (2009) | Northern Gaspé Peninsula, Québec, Canada | 689 | 1214 x-sec | 12 | ~30 km | 2540 |
| Butler and Sawyer (2008) | Lewis Range, Glacier National Park, Montana, USA | 22 | 22 x-sec | 2 | ~5 km | Not provided |
| Casteller et al. (2007) | Grisons, Switzerland | 145 | 122 x-sec 52 cores 10 wedges | 2 | ~ 20 km | Not provided |
| Germain et al. (2005) | Northern Gaspé Peninsula, Québec, Canada | 142 | 142 x-sec | 5 | *NA* | 420 |
| Dube et al.(2004) | Northern Gaspé Peninsula, Québec, Canada | 110 | 170 x-sec | 3 | ~9 km | Not provided |
| Hebertson and Jenkins (2003) | Wasatch Plateau, Utah, USA | 261 | Not provided | 16 | *NA* | Not provided |
| Rayback (1998) | Front Range, Colorado, USA | 98 | 58 trees cored (2-5 cores /tree) 31 x-sec 9 wedges | 2 | ~7 km | Not provided |
| Bryant et al. (1989) | Huerfano Valley, Colorado, USA | 180 | Not provided | 3 | ~2 km | Not provided |
| Butler and Malanson (1985a) | Lewis Range, Glacier National Park, Montana, USA | 78 | Not provided | 2 | ~6 km | Not provided |
| Butler (1979)~ | Glacier National Park, Montana, USA | NA | 36 x-sec 17 cores | 12 | ~15 km | Not provided |
| Smith (1973) | North Cascades, Washington, USA | NA | Not provided | 11 | ~ 35 km | Not provided |
| Potter (1969) | Absaroka Mountains, Wyoming, USA | 50 | Not provided | 5 | ~ 2 km | 50 |






### 1.2 Framework and objectives

Tree-ring avalanche research is resource and time intensive. Like other scientific fields, it is not feasible to completely sample the variable of interest with infinite detail due to logistical and financial constraints (Skøien and Blöschl, 2006). Thus, a strategic spatial sampling method is necessary. Here, we strategically sampled 12 avalanche paths in four distinct sub-regions of the U.S. northern Rocky Mountains of northwest Montana to examine spatial differences at a regional scale. The sampling strategy is based on the concept of scale triplet, which defines the spacing, extent, and support of our sampling scheme (Blöschl and Sivapalan, 1995). Incorporating the scale triplet concept helps us understand the nature of the problem, the scale at which measurements should be made, and how we can estimate the measurements across space. Often, the scale at which samples are collected differs from the scale necessary for predictive purposes (Blöschl, 1999). For example, if we are interested in avalanche frequency relationships with regional climate patterns but tree-ring samples are collected at an avalanche path scale, then a network of sampled paths need to be spaced and aggregated across the core of the climatically similar region. In our study, the extent is the entire region and sub-regions, the spacing is the distance between avalanche paths and sub-regions, and the support is the size of the area being sampled. In addition, the process scale is the natural variability of avalanche frequency, the measurement scale is the tree-ring proxies used to represent avalanche occurrence on an annual temporal scale, and the model scale relates to aggregating all of the sample areas to derive a regional avalanche chronology.

We adopt Germain's (2016) definition that large magnitude avalanches are events characterized by low and variable frequency with a high capacity for destruction. This generally translates to a size 3 or greater on the destructive classification scale - i.e. ability to bury or destroy a car, damage a truck, destroy a wood frame house, or break a few trees (Greene et al., 2016).

Understanding the spatiotemporal behavior of large magnitude avalanches on the regional scale will improve avalanche forecasting efforts, especially for operations involving avalanche terrain that impacts transportation corridors. Here, we aim to answer three specific questions:

1) What is the regional, sub-regional, and path specific frequency of large magnitude avalanches in the U.S. northern Rocky Mountains of northwest Montana?

2) How does the spatial extent of the study region affect the resultant avalanche chronology?

3) What is the probability of detecting regional avalanche activity by sampling different avalanche paths?

To our knowledge, this is the first study to look at how various spatial scales compare when reconstructing a regional avalanche chronology from dendrochronological data on a large dataset (N > 600 samples). Further, we believe this is the first study that utilizes a regional dendrochronological record to derive return periods over a large (> 3000 km$^2$) spatial extent. Our hypothesis is that aggregating the paths into sub-region and then again into a full region allows us to minimize the limitation of tree-ring avalanche chronologies underestimating avalanche years at these scales.



## 2. Methodology

### 2.1 Study Site

Our study site consists of 12 avalanche paths in the Rocky Mountains of northwest Montana, USA (Figure 1 and Table 2). We sampled sets of three avalanche paths in four distinct sub-regions within three mountain ranges: the Whitefish Range (WF, Red Meadow Creek) and Swan Range (Swan, Lost Johnny Creek) on the Flathead National Forest, and two sub-regions within the Lewis Range in Glacier National Park (GNP), Montana. The sites in GNP are along two major transportation corridors through the park: the Going-to-the-Sun Road (GTSR) and U.S. Highway 2 in John F. Stevens (JFS) Canyon. These two areas were utilized for previous dendrochronological avalanche research (Butler and Malanson, 1985a; Butler and Malanson, 1985b; Butler and Sawyer, 2008; Reardon et al., 2008). A robust regional avalanche chronology reconstruction will help place the previous work in context of the wider region. The other two sites, WF and Swan, are popular backcountry recreation areas with access via snowmachine in the winter along a U.S. Forest Service road. The avalanche paths in each sub-region encompass a range of spatial extents from adjacent (i.e. < 30 m apart) to ~10 km apart. Overall, this study region provides an ideal natural setting for studying avalanches due to its geography, inclusion of transportation and recreation corridors potentially impacted by avalanches, relative accessibility, and no artificial avalanche hazard mitigation.

**Figure 1: Study site. The red rectangle in the state of Montana designates the general area of the four sampling sites. The sites are (A) Red Meadow, Whitefish Range (WF), (B) Going-to-the-Sun Road (GTSR), central GNP, (C) Lost Johnny Creek, northern Swan Range (Swan), and (D) John F. Stevens Canyon (JFS), southern GNP. Satellite and map imagery: © Google (n.d.). Maps produced using ggmap in R (Korpela et al. 2019).**



**Table 2: Topographic characteristics of all avalanche paths. Different colors indicate sub-regions as shown in**
**Figure 1. * denotes two major starting zones for one runout in Shed 10-7 and Shed 7 paths.**

| Path | n | Full Path Elev. (mean) (m) | Full Path Elev. (range) (m) | Starting Zone Elev. (mean) (m) | Full Path Slope (mean) (°) | Starting Zone Slope (mean) (°) | Median Aspect (°) | Area (km²) | Length (m) | Vertical (m) | Years of previous fire or logging |
|---|---|---|---|---|---|---|---|---|---|---|---|
| WF-Red Meadow A (RMA) | 41 | 1651 | 1462 - 1957 | 1774 | 26 | 32 | 155 | 0.32 | 1004.97 | 495.20 | 1952 |
| WF-Red Meadow B (RMB) | 40 | 1870 | 1643 - 2164 | 1965 | 31 | 37 | 53 | 0.13 | 1041.98 | 521.27 | 1967 |
| WF-Red Meadow C (RMC) | 42 | 1650 | 1582 - 1742 | 1692 | 28 | 33 | 257 | 0.08 | 326.14 | 160.46 | 1962 |
| GTSR - 54-3 | 56 | 1501 | 1080 - 2149 | 1708 | 31 | 40 | 327 | 0.44 | 2063.61 | 1068.49 | NA |
| GTSR- Little Granite (LGP) | 109 | 1770 | 1109 - 2314 | 2170 | 24 | 34 | 250 | 0.78 | 2940.29 | 1205.07 | NA |
| GTSR- Jackson Glacier Overlook (JGO) | 41 | 1863 | 1500 - 2660 | 2090 | 32 | 42 | 180 | 0.70 | 1793.13 | 1159.84 | NA |
| Swan-Lost Johnny A (LJA) | 53 | 1619 | 1441 - 1896 | 1731 | 29 | 38 | 77 | 0.41 | 811.50 | 455.27 | 1971-72 |
| Swan-Lost Johnny B (LJB) | 26 | 1633 | 1478 - 1879 | 1721 | 32 | 39 | 76 | 0.57 | 617.52 | 401.80 | 1971-72 |
| Swan-Lost Johnny C (LJC) | 42 | 1550 | 1344 - 1750 | 1670 | 34 | 36 | 326 | 0.39 | 667.88 | 405.66 | 1957, 2003 |
| JFS-Shed 10-7 (S10.7)* | 109 | 1644 | 1233 - 2193 | 1910 1964 | 31 | 35 39 | 176 | 0.13 | 1745.66 | 959.74 | 1910 |
| JFS-Shed 7 (S7)* | 46 | 1712 | 1310 - 2078 | 1935 1837 | 29 | 34 36 | 152 | 0.57 | 1686.96 | 768.01 | 1910 |
| JFS-1163 | 50 | 1718 | 1250 - 2217 | 1861 | 38 | 42 | 158 | 0.17 | 1636.52 | 966.82 | 1910 |
| All Paths | 655 | 1690 | 1080 - 2660 | 1869 | -0.17 | 0.14 | 31 | 37 | Spatial footprint = 3500 km² | | |





Northwest Montana's avalanche climate is classified as both a coastal transition and intermountain avalanche
climate (Mock and Birkeland, 2000), but it can exhibit characteristics of both continental or coastal climates.
The elevation of avalanche paths within the study sites range from approximately 1100 m to 2700 m and the
starting zones of these paths are distributed among all aspects (Table 2).
We eliminated or minimized influence from exogenous disturbance factors such as logging and wildfire by
referencing wildfire maps extending back to the mid-20[th] century. We selected sites undisturbed by wildfire
since this time except for Lost Johnny Creek, which was purposeful as this area burned most recently in 2003.
We also minimized the influence of logging by selecting sites not previously logged. Using historical logging
parcel spatial data, we determined logging in some sites was limited to very small parcels adjacent to the
farthest extent of the runout zones.
The historical observational record in this area is limited. In this study region, the Flathead Avalanche Center
(FAC), a regional U.S. Forest Service backcountry avalanche center, records all avalanches observed and
reported to the center. However, not all avalanches are observed or reported given the approximately 3500
km$^2$ advisory area. The Burlington-Northern Santa Fe Railway (BNSF) Avalanche Safety Program records
most avalanches observed in John F. Stevens Canyon in southern Glacier National Park, where there is 16
km of rail line with over 40 avalanche paths. However, systematic operational observations only began in
2005. Observations prior to this time are inconsistent, though large magnitude avalanches were mostly
recorded. Reardon et al. (2008) developed as complete a record as possible from the Department of
Transportation and railroad company records, National Park Service ranger logs, and popular media archives.
In this sub-region avalanche mitigation is conducted on an infrequent and inconsistent basis in emergency
situations, which is typically only once a year, if at all. Thus, the record approximates a natural avalanche
record. We compared the reconstructed avalanche chronology of the JFS sub-region to the historical record
for qualitative purposes of large magnitude years. A quantitative comparison would not be reflective of the
true reliability of tree-ring methods because of the incomplete historical record.
**2.2 Sample Collection and Processing**
Our sampling strategy targeted an even number of samples collected from both lateral trimlines at varying
elevations and trees located in the main lower track and runout zone of the selected avalanche paths. This
adequately captured trees that were destroyed and transported, as well as those that remained in place.
Sample size for avalanche reconstruction using tree-ring data requires careful consideration. Butler and
Sawyer (2008) suggest that a few damaged trees may be sufficient for avalanche chronologies, but larger
target sample sizes increase the probability of detecting avalanche events (Corona et al., 2012). Germain et
al. (2010) examined cumulative distribution functions of avalanche chronologies and reported only slight
increases in the probability of extending chronologies with sample size greater than 40. Thus, given the large
spatial footprint (~3500 km$^2$) of this study and feasibility of such a large sample size, we sampled between
26-109 samples per avalanche path resulting in 655 trees (Table 2). Eight trees were unsuitable for analysis
leaving us with 673 total samples from 647 trees. Of the 673 total samples, we collected 614 cross sections




and 59 cores. Shed 10.7 (S10.7) path was the focus of previous work (Reardon et al., 2008), and the
dendrochronological record extends up to 2005 (n=109 trees). Little Granite Path (LGP) was collected in the
summer of 2009 (n=109 trees). We sampled the remaining 10 paths (437 of the 655 total trees) in the summer
of 2017.
We collected full stem cross-sections from dead (both downed and standing dead) trees, and cores from live
trees. We used predominantly cross-sections in this study for a more robust analysis as events can potentially
be missed or incorrectly identified in cores. We emphasized the selection of trees with obvious external scars
and considered location, size, and potential age of tree samples. A limitation of all avalanche
dendrochronology studies is that large magnitude events cause extensive damage and high tree mortality,
thereby reducing subsequent potential tree-ring records.
We sampled stem cross-sections at the location of an external scar or just above the root buttress from downed
or standing and dead trees, and from stumps of trees topped by avalanche damage. We extracted tree-ring
core samples from live trees with obvious scarring or flagging along the avalanche path margins and runout
zone using a 5 mm diameter increment borer. We collected a minimum of two and up to four core samples
per tree (two in the uphill-downhill direction and two perpendicular to the slope). We photographed each
sample at each location and recorded species, GPS coordinates (accuracy 1-3 m), amount of scarring on the
cambium of the tree, relative location of the tree in the path, and upslope direction (Peitzsch et al., 2019). We
also recorded location characteristics that identified the tree to be in-place vs. transported from its original
growth position (i.e. presence or absence of roots attached to the ground or the distance from an obvious
excavated area where the tree was uprooted).
To prevent radial cracking and further rot, we dried and stabilized the cross sections with a canvas backing.
We sanded samples using a progressively finer grit of sandpaper to expose the anatomy of each growth ring,
and used the visual skeleton plot method to account for missing and false-rings and for accurate calendar
year dating (Stokes and Smiley, 1996). We assessed cross-dating calendar-year accuracy of each sample and
statistically verified against measured samples taken from trees within the gallery forest outside the avalanche
path, and from preexisting regional chronologies (Table A1) (ITRDB, 2018) using the dating quality control
software COFECHA (Grissino-Mayer, 2001;Holmes, 1983). For further details on cross-dating methods and
accuracy calculation for this dataset see Peitzsch et al. (2019).
**2.3 Avalanche Event Identification**
We analyzed samples for signs of traumatic impact events (hereafter "responses") likely caused by snow
avalanches. We adapted a classification system from previous dendrogeomorphological studies to
qualitatively rank the trauma severity and tree growth response from avalanche impacts using numerical
scores ranked 1 through 5 (Reardon et al., 2008). This classification scheme identified more prominent
avalanche damage responses with higher quality scores, and allowed us to remain consistent with previous
work (Corona et al., 2012; Favillier et al., 2018) (Table 3). To compare our ability to capture avalanche /
trauma events using cores versus those captured using cross-sections, we sampled a subset (n=40) of the




cross-sections by analyzing four 5 mm wide rectangles to mimic a core sample from an increment borer. The
four subsamples on each cross section were made perpendicular to one another (i.e. 90 degrees) based on the
first sample taken from the uphill direction of each stem to replicate common field sampling methods. We
then summarized results from the four subsamples for each tree by taking the highest response score for each
growth year. Finally, we compared the number, quality response category, and calendar year of the avalanche
/ trauma events derived from the core subsamples to those identified from the full cross sections.

**Table 3: Avalanche impact trauma classification ratings.**

| Classification | Description |
|---|---|
| $C_1$ | • Clear impact scar associated with well-defined reaction wood, growth suppression or major traumatic resin duct development.<br>• Or, the strong presence of some combination of these major anatomical markers of trauma and growth response recorded in multiple years of growth and occurring at a year that multiple samples from other trees at the site record similar trauma and scaring.<br>• $C_1$ events are also assigned to the death date of trees killed by observed avalanche mortality at the collection site; the presence of earlywood indicates an early spring, or late avalanche season, event killed the tree. |
| $C_2$ | • Scar or small scar recorded in the first ten years of tree growth without associated reaction wood, growth suppression or traumatic resin ducts.<br>• Or, obvious reaction wood, growth suppression or significant presence of traumatic resin ducts that occur abruptly after normal growth that lasts for 3 or more years. |
| $C_3$ | • The presence of reaction wood, growth suppression, or traumatic resin ducts recorded in less than 3 successive growth years. |
| $C_4$ | • Poorly defined reaction wood, growth suppression or minimal presence of traumatic resin ducts lasting 1-2 years.<br>• Or, a $C_3$ class event occurring in the first 10 years of tree growth where the cause of damage could result from various biological and environmental conditions. |
| $C_5$ | • Very poorly defined reaction wood, growth suppression, or minimal presence of traumatic resin ducts isolated in one growth year.<br>• Or, a $C_4$ class event occurring in the first 10 years of tree growth where the cause of damage could result from various biological and environmental conditions. |



**2.4 Chronology and Return Period Calculation**
To generate avalanche event chronologies and estimate return periods for each path and for the entire study
site, we utilized *R* statistical software and the package *slideRun*, an extension of the *burnR* library for forest
fire history data (Malevich et al., 2018). We calculated the age of each tree sampled and the number of
responses per year and computed descriptive statistics for the entire dataset. Estimates of avalanche path
return intervals should be viewed as maximum return interval values due to the successive loss of samples
and decreasing sample number back through time.
We used a multi-step process to reconstruct avalanche chronologies on three different spatial scales:
individual paths, four sub-regions, and the entire region. We also calculated a regional avalanche activity
index (RAAI) (Figure 2). The process involved first calculating the ratio of trees exhibiting GD over the
number of samples alive at year *t* to provide the index $I_t$ (Shroder, 1978):

$$I_t = \left( \frac{\sum_{i=1}^{n}(R_t)}{\sum_{i=1}^{n}(A_t)} \right) \times 100 \tag{1}$$

where *R* is the number of trees recording a GD at year *t* with $A_t$ representing the number of trees alive in our
samples at year *t*.

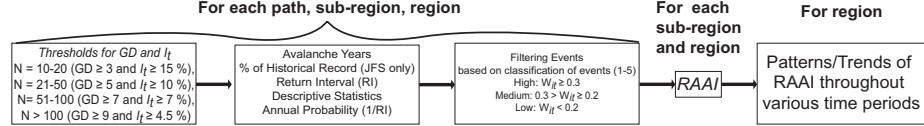

**Figure 2: General workflow of analytical methods to reconstruct regional avalanche chronology and regional**
**avalanche activity index.**
We then used double thresholds to estimate the minimum absolute number of GD and a minimum percentage
of samples exhibiting GD per year ($I_t$) based on sample size following thresholds established by Corona et
al. (2012) and Favillier et al. (2018): N = 10-20 (GD ≥ 3 and $I_t$ ≥ 15 %), N = 21-50 (GD ≥ 5 and $I_t$ ≥ 10 %),
N= 51-100 (GD ≥ 7 and $I_t$ ≥ 7 %), and N > 100 (GD ≥ 9 and $I_t$ ≥ 4.5 %). We then estimated the number of
avalanche years, descriptive statistics for return intervals (RI), and the annual probability (1/RI) for each
path, sub-region, and region.
We used the chronologies derived from this process to calculate a weighted index factor ($W_{it}$). We used this
established threshold approach since it has been broadly employed in the literature and allows comparability
of our avalanche chronology to results reported in other studies. We adapted previous equations of a weighted
response index (Kogelnig-Mayer et al., 2011) to our 5-scale ranking quality classification to derive the $W_{it}$:

$$W_{it} = \left( \left( \sum_{i=1}^{n} T_{C_1} * 7 \right) + \left( \sum_{i=1}^{n} T_{C_2} * 5 \right) + \left( \sum_{i=1}^{n} T_{C_3} * 3 \right) + \left( \sum_{i=1}^{n} T_{C_4,C_5} \right) \right) * \frac{\sum_{i=1}^{n} R_t}{\sum_{i=1}^{n} A_t} \tag{2}$$

where the sum of trees with scars or injuries ($C_1$ - $C_5$) were multiplied by a factor of 7, 5, 3, 1 and 1
respectively (Kogelnig-Mayer et al., 2011).





Next, we classified $W_{it}$ into high, medium, and low confidence events using the thresholds detailed in Favillier
et al. (2018), where High: $W_{it} \geq 0.3$, Medium: $0.3 > W_{it} \geq 0.2$, Low: $W_{it} < 0.2$. This provided another step
discriminating the avalanche events/years signal from noise. We included all events with medium to high
confidence in the next analysis. For this subset of higher-quality events, we calculated the number of
avalanche years, descriptive statistics for return intervals, and annual probability for each avalanche path,
sub-region, and overall region. We then compared return intervals for all individual paths and sub-regions
using analysis of variance (ANOVA) and Tukey's Honest Significant Difference (HSD) (Ott and
Longnecker, 2016).
Next, we compared the number of avalanche years and return periods identified in the full regional
chronology to subsets of the region to determine the number of paths required to replicate a full 12-path
regional chronology. We assessed the full chronology against a subsampling of 11 total paths by sequentially
removing the three paths with the greatest sample size. We then randomly sampled two paths from each sub-
region for a total subsample of eight paths, followed by generating a subsample of four paths by choosing
the path in each sub-region with the greatest sample size. Finally, we selected a random sample of one path
from each sub-region to compare against a total of four single path subsamples.

**2.5 Regional Avalanche Activity Index and Probability of Detection**

Next, we used the $I_t$ statistic from each path to calculate a regional avalanche activity index (RAAI) for the
sub-regions and overall region (Germain et al., 2009). The RAAI for each year across the sub-regions and
region provides a more comprehensive assessment of avalanche activity within the spatial extent. For each
year $t$, we calculated RAAI:

$$RAAI_t = \left( \sum_{i=1}^{n} I_t \right) \Big/ \left( \sum_{i=1}^{n} P_t \right) \qquad (3)$$

where $I$ is the index factor as per Eq. (1) for a given avalanche path for year $t$ and $P$ is the number of paths
that could potentially record an avalanche for year $t$. For the calculation of the overall RAAI, we required
each path to retain a minimum sample size of $\geq 10$ trees with a minimum number of three paths for year $t$,
and a minimum of one path from each sub-region. We performed a sensitivity test to establish the minimum
number of paths necessary to calculate an RAAI value for any given year.
We also calculated the probability of detecting an avalanche year identified in the regional chronology as if
any given individual path was sampled. The probability of detection for a given year ($POD_{year}$) is defined as:

$$POD_{year} = \frac{a}{a+b} \qquad (4)$$

where $a$ is the number of individual avalanche paths that identify any given avalanche year in the regional
chronology and $b$ is the total number of avalanche paths (n=12). We calculated $POD_{year}$ for every year in the
regional avalanche chronology. We then compared the $POD_{year}$ of individual paths to the number of active
avalanche paths as defined in Eq. (3).
We also calculated the probability of detection for each path for the period of record ($POD_{path}$):



$$POD_{path} = \frac{c}{c + d} \qquad (4)$$

where $c$ is the number of years identified in any given path that is included in the regional chronology and $d$
is the number of years in the regional chronology that are not identified in the chronology for the given path.
Finally, we examined trends in the RAAI through time using the non-parametric modified Mann-Kendall test
for trend (Mann, 1945;Hamed and Rao, 1998). We parsed the dataset into four periods to allow comparability
due to the loss of evidence and a decreasing sample size going back in time: the entire period of record, 1933
to 2017, 1950 to 2017, and 1990 to 2017. We selected these time periods based on the years with greatest
responses and peak RAAI values (1933, 1950, and 1990). We excluded intervals after 1990 to retain a
minimum of ~30-year record.

### 2.6 Geomorphological characteristics

Using a 10 m digital elevation model (DEM), we calculated a number of geomorphological characteristics
for each path, including mean elevation (m, full path and starting zone), elevation range (m), eastness
($\sin(aspect)$) and northness ($\cos(aspect)$) (radians), slope (degrees, full path and starting zone), curvature
(index (0-1), profile and planform), roughness (index, full path and starting zone), perimeter ($km^2$), area
($km^2$), length (m), and vertical distance from starting zone to runout zone (m). We also calculated the mean
of these characteristics for all paths in the region. The geomorphological characteristics allowed for a
determination of the representativeness of the region as a whole (i.e. are the paths similar across the region?)
as well as a comparison of the return interval for each path relative to these characteristics. Finally, we
estimated the potential relationship between path length, starting zone slope angle, the number of avalanche
years, and median return interval for each individual path using the Pearson correlation coefficient.

### 3. Results

We collected a total of 673 samples from 647 suitable avalanche impacted or killed trees in the full 12-path
regional avalanche collection. Of those 673 samples, 614 were cross sections (91%) and 59 were cores (9%).
Within these samples we identified 2134 GD, of which 1279 were classified as $C_1$ and $C_2$ (60%) (Figure 3(a
and b)). The oldest individual tree sampled was 367 years, and the mean age of all samples was 73 years
(Figure 3(c)). The period of record of sampled trees extended from 1636 to 2017 C.E. The most common
species in our dataset was *Abies lasciocarpa* (*ABLA*, sub-alpine fir) (46%) followed by *Pseudotsuga menziesii*
(*PSME*, Douglas-fir) (37%) and *Picea engelmannii* (*PIEN*, Engelmann spruce) (14%) (Figure 3(d)). The
oldest GD response dates to year 1655. In the entire dataset, the five years with the greatest number of raw
GD responses were 2002 (165 responses), 2014 (151 responses), 1990 (93 responses), 1993 (90 responses),
and 1982 (75 responses).


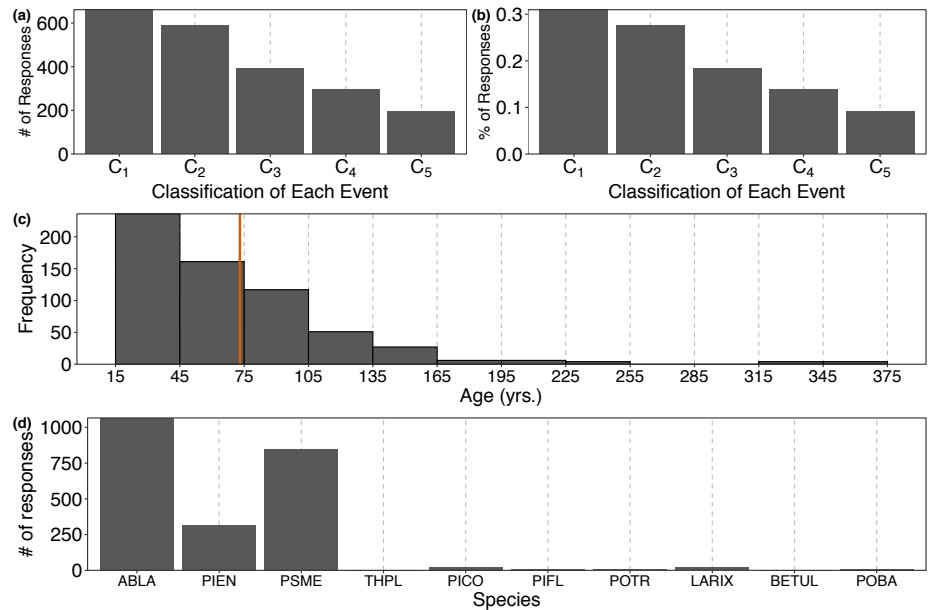

**Figure 3: Histograms of (a) number of classification of responses, (b) percentage of classification of responses, (c) sample age (red line represents mean age), and (d) species. For species: ABLA=*Abies Lasciocarpa*, PIEN = *Picea engelmannii*, PSME = *Pseudotsuga menziesii*, THPL = *Thuja plicata*, PICO = *Pinus contorta*, POTR = *Populus tremuloides*, LARIX = *Larix* Mill., BETUL = *Betul* L., POBA = *Populus balsamifera*.**

### 3.1 Avalanche Event Detection: Cores versus Cross-Sections

The avalanche event subset analysis that compared results obtained as if samples were from cores versus full cross sections showed that core samples alone would have missed numerous avalanche events and generated a greater proportion of low-quality growth disturbance classifications (Figure 4). For the subset of 40 samples analyzed as cores we identified only 124 of 191 (65%) total GD. Of the 66 GDs that we would have missed just by using cores, 24 were classified as $C_1$ quality events, 24 were $C_2$, 14 were $C_3$, 3 were $C_4$, and 2 were $C_5$.
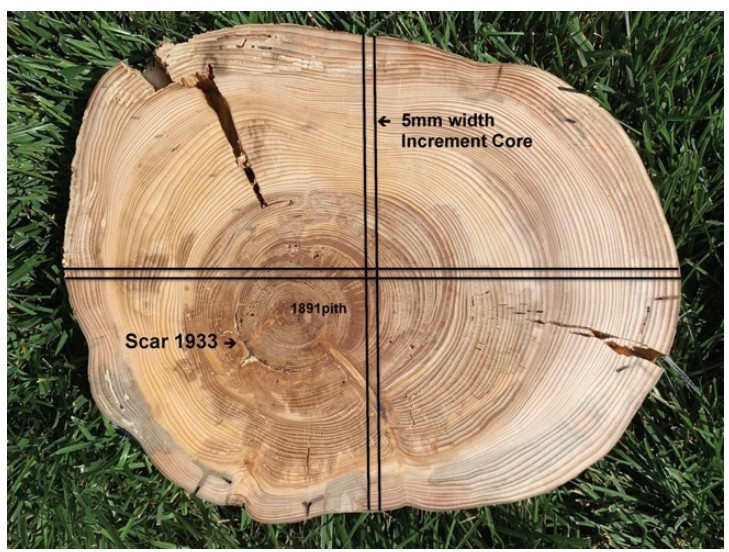

311

**Figure 4: Example of cross section sample where 4 cores taken on uphill, downhill, and perpendicular (2) would have missed at least one scar (1933) and potentially the pith of the tree. The black lines indicate the potential cores using a 5 mm width increment borer.**

**3.2 Individual Path Chronologies**

There were 49 avalanche events identified from GD responses across all 12 individual paths in the study region. The avalanche years most common throughout all of the individual path chronologies were: 2014 (7 paths); 1982 and 1990 (5 paths); and 1933, 1950, 1972, and 1974 (4 paths) (Figure 5 and Table 4). We identified the year with the greatest number of individual tree GD responses (2002) in 3 paths. Two of these paths were in the JFS sub-region as well as the RMA path in the WF sub-region. There was no clear pattern of paths physically closer in proximity to each other having more similarly identified avalanche years. However, paths within the WF sub-region produced the most similar number of large magnitude avalanche years. When we applied the $W_{it}$ process step to more heavily weight higher quality signals, the number of identified avalanche years did not change for any individual avalanche path compared to application of the double threshold method alone. This highlights the number of responses classified as $C_1$ and $C_2$ (high quality) in our dataset.



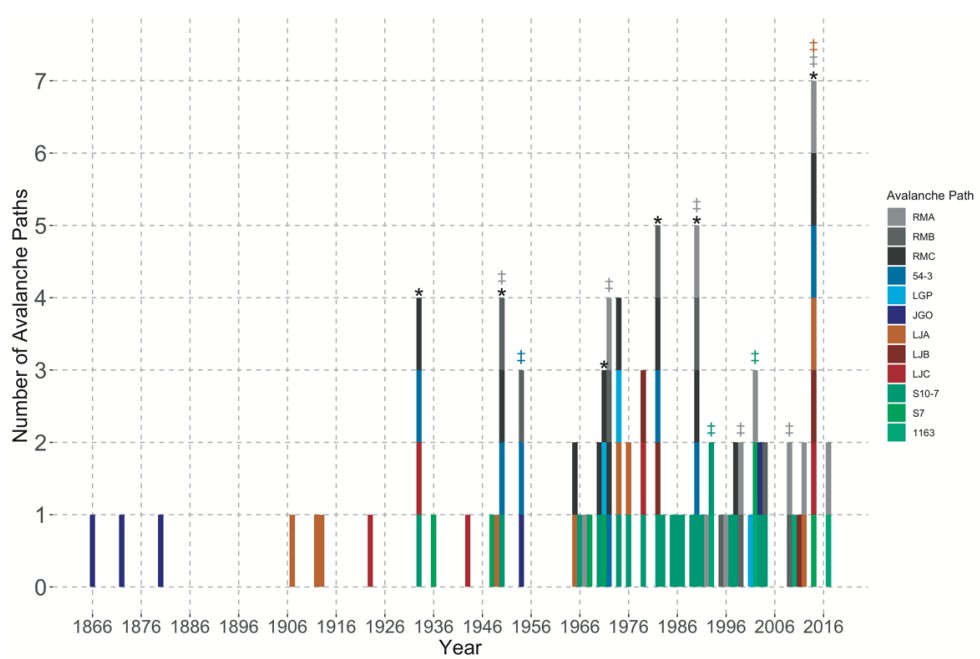


**Figure 5: Number of individual avalanche paths identified per year. Avalanche years with ‡ (gray=WF, dark blue = GTSR, orange = Swan, green= JFS) indicate years identified in at least two avalanche paths in the sub-region. * represents avalanche years in common in at least 1 path from at least three of the four sub-regions.**



**Table 4: Avalanche chronologies and return interval (RI) statistics of all 12 avalanche paths in the region.**
**Avalanche years in bold indicate years identified in at least two avalanche paths in the sub-region. Underlined**
**avalanche years indicate years in common in at least 1 path from at least three of the four sub-regions.**

|  | RMA | RMB | RMC | 54-3 | LGP | JGO | LJA | LJB | LJC | Shed 10-7 | Shed 7 | 1163 |
|---|---|---|---|---|---|---|---|---|---|---|---|---|
| **Aval Years** | 1967 **1972** **1990** 1992 1996 **1999** 2002 **2009** 2012 **2014** 2017 | **1950** 1954 **1972** **1982** **1990** 1995 **1999** 2004 **2009** | 1933 **1950** 1965 1970 **1971** **1972** 1974 1982 **1990** 1998 **2014** | 1933 1950 **1954** 1972 1982 1990 2014 | 1971 | 1866 1872 1880 2001 2009 | 1907 1912 1913 1949 1965 1974 **1954** 1976 2012 **2014** | 1979 1982 2011 **2014** | 1923 1933 1943 1979 **2014** | 1933 1950 1966 1970 1974 1976 1979 1982 1983 1985 1986 1987 1989 1990 1991 **1993** 1997 1998 2003 2004 | 1936 1948 1968 1971 **2002** 2014 | **1993** **2002** 2010 2017 |
| **# of aval. years** | 11 | 9 | 11 | 7 | 4 | 5 | 9 | 4 | 5 | 20 | 6 | 4 |
| **RI - median** | 3 | 5 | 8 | 14 | 8 | 8 | 7 | 3 | 22.5 | 2 | 12 | 8 |
| **RI - mean** | 5 | 7.38 | 8.1 | 13.5 | 12.67 | 34.25 | 13.38 | 11.67 | 22.75 | 3.74 | 15.6 | 8 |
| **RI – min.** | 2 | 4 | 1 | 4 | 3 | 6 | 1 | 3 | 10 | 1 | 3 | 7 |
| **RI – max.** | 18 | 18 | 17 | 24 | 27 | 74 | 36 | 29 | 36 | 17 | 31 | 9 |
| **1/RI** | 0.33 | 0.20 | 0.13 | 0.07 | 0.13 | 0.13 | 0.14 | 0.33 | 0.04 | 0.50 | 0.08 | 0.13 |
| **σ** | 4.81 | 4.78 | 6.12 | 7.42 | 12.66 | 33.09 | 14.79 | 15.01 | 14.73 | 4.68 | 10.50 | 1.00 |


Across all individual paths, the median estimated return interval was 8 years with a range of 2 to 22.5 (Figure
6). JGO, located in the GTSR sub-region, exhibited the greatest spread in estimated return intervals followed
by LJB. The avalanche paths within the GTSR sub-region had the most similar return intervals of any of the
sub-regions whereas the paths in the JFS sub-region exhibited substantial variability in median return interval
values. The return interval for JGO differed significantly from several other paths: RMA, RMB, RMC, and
Shed 10-7 ($p \leq 0.01$). However, when we relax a strict cutoff of $p = 0.05$, the return interval from JGO also
differed from 1163 ($p = 0.07$) and LJA ($p = 0.08$). Similarly, the return interval for Shed 10-7 differs from LJC
($p = 0.07$). In assessing the potential geomorphic controls on return interval, path length was the only
significantly correlated characteristic ($r = 0.65$, $p = 0.02$, Figure A1).


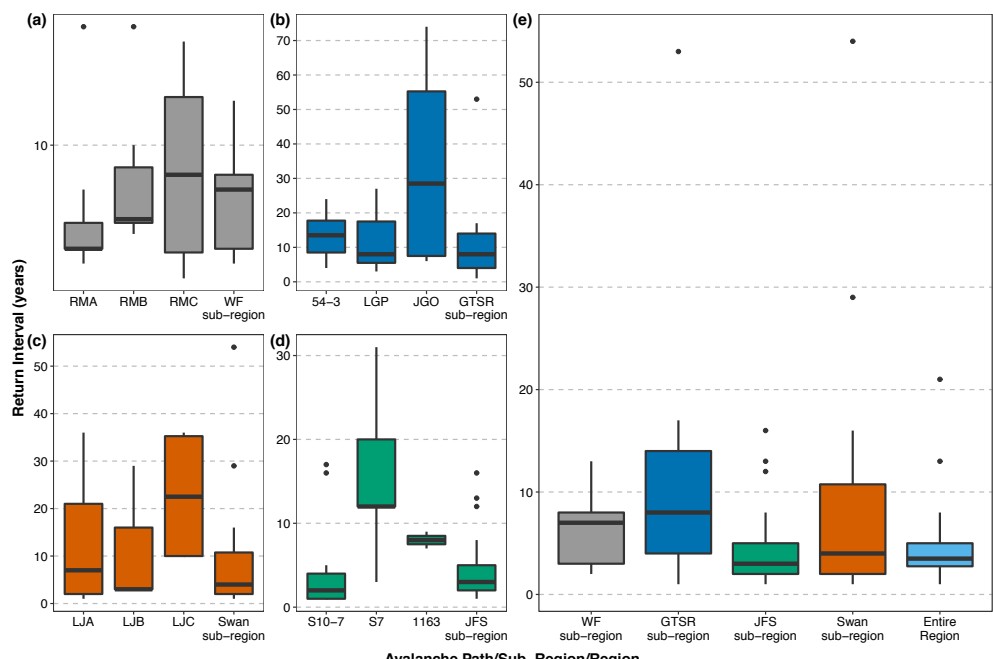


**Figure 6: Boxplot of return intervals for individual avalanche paths in each sub-region: (a) WF, (b) GTSR, (c) Swan, and (d) JFS. (e) shows the median return intervals for the sub-regions and the overall region.**

### 3.2 Sub-region Chronologies

When the paths were aggregated into sub-regions (three paths per sub-region) the median return periods for each sub-region were similar and all less than 10 years (Figure 6(e) and Table 5). The number of avalanche years for all of the sub-regions ranges from 12-18 with the greatest number of identified years in the JFS sub-region and the fewest in the WF sub-region. The JFS sub-region has the shortest median return interval followed by the Swan, WF, and GTSR sub-regions. The number of avalanche years for each aggregated sub-region is greater than the number of avalanche years for any individual path within each sub-region except for the JFS sub-region where 18 avalanche years were identified but Shed 10-7 totaled 20 avalanche years (Table 6).





**Table 5: Avalanche chronologies and return interval (RI) statistics of all four sub-regions.**

|  | WF | GTSR | Swan | JFS | Region |
|---|---|---|---|---|---|
| **# of aval. years** | 12 | 14 | 13 | 18 | 30 |
| **RI – median** | 7 | 8 | 4 | 3 | 3 |
| **RI – mean** | 6.27 | 11.35 | 11.25 | 4.94 | 5.21 |
| **RI – min.** | 2 | 1 | 1 | 1 | 1 |
| **RI – max.** | 13 | 53 | 54 | 16 | 53 |
| **1/RI** | 0.14 | 0.13 | 0.25 | 0.33 | 0.33 |
| **σ** | 3.69 | 13.48 | 15.70 | 4.60 | 9.53 |

**Table 6: Number of avalanche events for each subregion, the mean of three individual paths in each region, and**
**the overall aggregated region.**

| # of avalanche events | | |
|---|---|---|
| **Sub-region** | **3 individual paths** | **Aggregated sub-region** |
| WF | 11,9,11 | 12 |
| GTSR | 7,3,5 | 14 |
| Swan | 9,4,5 | 13 |
| JFS | 20,6,4 | 18 |
| **Region** | 30 | |


In terms of commonality of years between the sub-regions, 1982 is the only year identified in all of the four
sub-regions (Figure 7). Avalanche years commonly identified in three sub-regions are 1950, 1954, 1974 and
2014. The JFS sub-region identified the greatest number of years exclusive to that sub-region (10 years). The
WF sub-region shared the greatest number of years with other regions (11 years) followed by JFS (9 years),
GTSR (8 years), and the Swan (7 years). In the only available comparison against an incomplete and limited
historical record, the individual reconstructed avalanche chronologies of paths in the JFS sub-region captured
10-50% of the recorded large magnitude events over years 1908 to 2017.
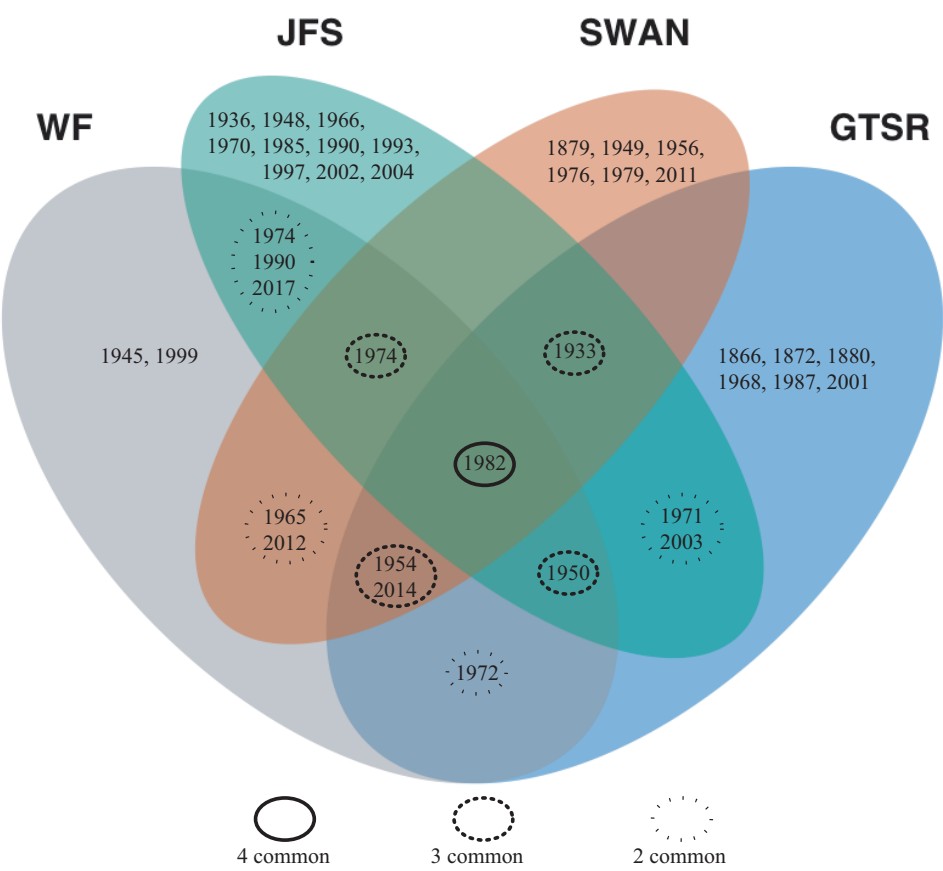

**Figure 7: Venn diagram of avalanche years common between sub-regions. Overlapping areas of each ellipse indicate years in common with each sub-region.**

**3.3 Regional Chronology and RAAI**

We identified 30 avalanche years in the overall region and a median return interval of 3 years (Table 5). The number of samples increases through time to a peak during 2005 and as expected the number of GD also increases through time (Figure 8(a)). The $W_{it}$ index also increases, particularly from year 2000 onward with the largest spikes in 2014 and 2017 (Figure 8(b)). The regional assessment of avalanche years identified fewer years (n=30) than the simple aggregation of all unique avalanche years identified in the individual paths (n=49) (Table A2).



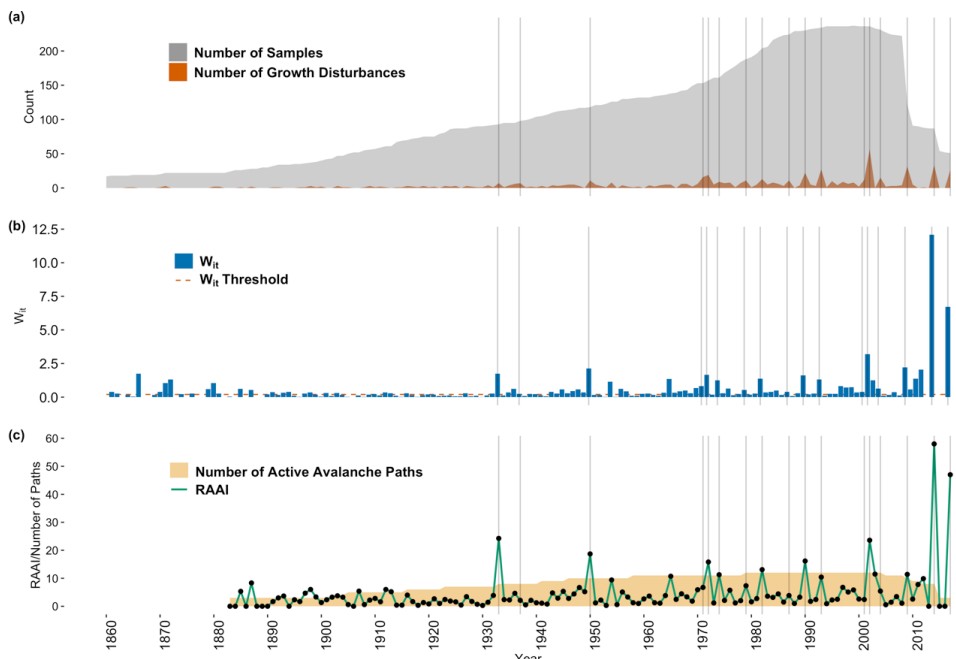

**Figure 8: (a)** The number of samples (gray shaded area) increases through time, but the number of responses (dark orange shaded area) varies. **(b)** The $W_{it}$ threshold (0.2, red dashed line) provides a means of discriminating between high and low confidence signals in the tree ring record. **(c)** The RAAI (green line, black points) is a measure of regional avalanche activity based on the $I_t$ of each path and the number of active avalanche paths (yellow shaded area).

When we included all paths but S10.7 (one of two paths with the greatest sample size), we captured 80% of all avalanche years and added one new year to the chronology (Table 7). When we removed LGP (the other path with the greatest sample size), we still captured all of the years in the regional chronology but introduced four new years into the chronology for a total of 34 years. A random sample of eight (two from each sub-region) of the 12 avalanche paths captured 83% of the years in the chronology and identified two new avalanche years. Finally, when using only one path from each sub-region with the largest samples size (Shed 10-7, 54-3, LJA, and RMA), we captured 73% of the avalanche years identified in the full regional chronology. When using a random sample of one path from each sub-region (1163, LGP, LJC, RMB), we captured only 43% of the years included in the regional chronology of all 12 paths. The RAAI is insensitive (no significant difference, p > 0.05) to the number of paths when tested using a minimum number of paths recording an avalanche in year *t*. The years with the largest RAAI are 2014 and 2017 followed by 2002, 1950 and 1933 (Figure 8(c)).





**Table 7: Comparison of the number of avalanche years and RI when including all 12 paths in region to using a**
**combination of fewer paths to define the region.**

| Paths | Region (All Paths) | All but S10.7 | All but LGP | All but 54-3 | S7, 1163, LGP, JGO, RMB, RMC, LJB, LJC | S10.7, 54-3, LJA, RMA | 1163, LGP, LJC, RMB |
|---|---|---|---|---|---|---|---|
| # Paths | 12 | 11 | 11 | 11 | 8 | 4 | 4 |
| Sample Size (n) | 635 | 528 | 526 | 581 | 382 | 253 | 239 |
| # of Aval Years | 30 | 27 | 34 | 31 | 27 | 34 | 17 |
| # matches with regional | NA | 24 | 30 | 29 | 25 | 22 | 13 |
| # not in regional | NA | 1 | 4 | 2 | 2 | 11 | 4 |
| % captured in regional | NA | 80 | 100 | 97 | 83 | 73 | 43 |
| Median RI | 3 | 3 | 3 | 3 | 3 | 2 | 3.5 |
| # years removed using only $W_{it=HLC}$ instead of $W_{it=MLC}$ and HLC | 10 | 3 | 9 | 7 | 1 | 1 | 1 |


To assess potential long-term trends in regional avalanche activity we implemented the modified Mann-
Kendall test since the chronology exhibits weak serial autocorrelation. The full period of record of RAAI
(1867-2017) exhibits a positive trend (tau = 0.186, Sens slope = -0.01, p = 0.006). The two other periods
analyzed, 1950-2017 and 1990-2017, exhibit neither a positive nor a negative trend (p = *0.*36 and p = 0.95,
respectively).
The probability of detection for the avalanche years ($POD_{year}$) identified in the regional chronology ranged
from 8 to 58% when we examined individual paths (Table 8). The year with the highest *POD* was 2014. The
mean *POD* for all years was 21%. When we examined avalanche paths that exhibited at least one scar during
avalanche years identified in the regional chronologies, the *POD* is generally greater.




**Table 8: Probability of Detection ($POD_{year}$). Avalanche years identified in the regional chronology and associated**
***POD* by analyzing individual paths with and without GD, sample size, and $W_{it}$ thresholds.**

| Avalanche Year in Regional Chronology | POD (%) with thresholds | POD (%) without thresholds |
|---|---|---|
| 1866 | 8 | 8 |
| 1872 | 8 | 8 |
| 1880 | 8 | 17 |
| 1933 | 33 | 58 |
| 1936 | 8 | 25 |
| 1945 | NA | 58 |
| 1948 | 8 | 33 |
| 1950 | 33 | 58 |
| 1954 | 25 | 67 |
| 1956 | NA | 58 |
| 1965 | 17 | 67 |
| 1970 | 17 | 50 |
| 1971 | 25 | 50 |
| 1972 | 33 | 83 |
| 1974 | 33 | 75 |
| 1976 | 17 | 50 |
| 1982 | 42 | 92 |
| 1990 | 42 | 83 |
| 1993 | 17 | 50 |
| 1997 | 8 | 92 |
| 1998 | 17 | 50 |
| 1999 | 17 | 58 |
| 2002 | 25 | 75 |
| 2003 | 17 | 33 |
| 2004 | 17 | 75 |
| 2009 | 17 | 33 |
| 2011 | 8 | 33 |
| 2012 | 17 | 42 |
| 2014 | 58 | 58 |
| 2017 | 17 | 25 |
| **Mean** | **21** | **52** |


Finally, the probability of capturing all of the avalanche years identified in the regional chronology by each
individual path ranges from 7% to 40% (Table 9). The greatest $POD_{path}$ value from any given path is S10.7
($POD = 40\%$) in the JFS sub-region followed by RMC in the Whitefish sub-region ($POD = 37\%$). In general,
the paths within the Whitefish sub-region capture the regional chronology most consistently.




**Table 9: Probability of Detection of each individual path (*POD$_{path}$*) to the regional avalanche chronology.**

| Path | POD (%) |
|---|---|
| RMA | 27 |
| RMB | 27 |
| RMC | 37 |
| 54-3 | 23 |
| LGP | 7 |
| JGO | 17 |
| LJA | 17 |
| LJB | 10 |
| LJC | 7 |
| Shed 10-7 | 40 |
| Shed 7 | 17 |
| 1163 | 10 |

**4. Discussion**
The processing and analysis of 673 samples spanning a large spatial extent allowed us to create a robust
regional large magnitude avalanche chronology reconstructed using dendrochronological methods. Cross-
sections provided a more robust and complete GD and avalanche chronology compared to a subsample
generated from cores alone. Due to the reduced information value of working only with cores, Favillier et al.
(2017) included a discriminatory step in their methods to distinguish avalanche signals in the tree-ring record
from exogenous factors, such as abnormal climate signals or response to insect disturbance. By using cross
sections to develop our avalanche chronologies, we were able to view the entire ring growth and potential
disturbance around the circumference of the tree as opposed to the limited view provided by cores. This
allowed us to place GD signals in context to both climate and insect disturbance without the need for this
processing step. We identified 2134 GD from our samples, which is similar to Martin and Germain's (2016)
study where they collected 458 cross sections and 350 cores, and reported 2251 GD.
We targeted sample collection in areas in the runout zones and along the trim line where large magnitude
avalanches occurred in recent years. However, at several sites we also collected samples up into the bottom
of the track (S10.7, Shed 7, and 1163). Thus, some additional noise in the final chronology for those specific
paths could be due to more frequent small magnitude avalanches. Though the oldest individual trees extended
as far back as the mid-17$^{th}$ century, the application of the double thresholds and $W_{it}$ processing steps restricted
individual path avalanche chronology lengths since the minimum GD threshold requirements were not met.
It is difficult to place much confidence in these older recorded events due to the decreasing evidence back in
time inherent in avalanche path tree-ring studies. Therefore, we chose to examine more recent time periods
with a larger sample size and more consistent number of sites and individual series when considering return
intervals and RAAI.
All of the paths in the study are capable of producing large magnitude avalanches with path lengths greater
than 100 m (typical length for avalanche destructive size 2, D2), and all but RMC have a typical path length
of close to or greater than 1000 m (for avalanche destructive size 3, D3) (Greene et al., 2016) . As Corona et





al. (2012) note, the avalanche event must be large enough to create an impact on the tree, and size D2 or
greater will be evident from the tree-ring record (Reardon et al., 2008). However, the successive damage and
removal of trees from events size D2 or greater also impacts the future potential to record subsequent events
of similar magnitude. In other words, if a large magnitude avalanche removes a large swath of trees in one
year, then there are fewer trees available to record a slightly smaller magnitude avalanche in subsequent
years. Therefore, dendrochronology methods inherently underestimate avalanche events by up to 60%
(Corona et al., 2012), and our results suggest these methods captured about 10-50% of the available historical
record for JFS canyon.

### 4.1 Regional Sampling Strategy

By examining three different spatial scales (individual path, sub-region, and region) we produced a large
magnitude avalanche chronology for the region captured in a small subset of the total number of paths across
the large region. Accordingly, this sampling strategy may also alleviate the issue of recording large magnitude
avalanches within a region in the successive years following a major destructive avalanche event that
removed large number of trees within specific paths but not others. Overall, a regional sampling strategy
enables us to capture large magnitude avalanche events over a broad spatial extent that is useful for regional
avalanche forecasting operations and future climate association analysis. This strategy also allows us to
understand large magnitude avalanche activity at scales smaller than the regional scale.

### 4.2 Chronologies for Individual Paths and Sub-Regions

We applied the $W_{it}$ threshold specifically to weight higher quality signals. The number of identified avalanche
years does not change for any individual avalanche path when we applied the $W_{it}$ process. This suggests that
many of the signals in our samples were ranked as high-quality (i.e. $C_1$-$C_2$). This can be attributed to the use
of cross sections which allowed for a more complete depiction and assessment of the tree-ring signal (Carrara,

469 1979).

We developed avalanche chronologies for 12 individual avalanche paths. The path with the greatest number
of identified avalanche years, S10.7, contains two major starting zones that are both steeper (35 and 39
degrees) than Shed 7, which also contains two separate starting zones. Reardon et al. (2008) collected a
substantial number of samples at higher elevations in the avalanche path. However, the location data for these
samples were not available. Many of those samples were the living stumps that captured smaller annual
events. This is likely the root of the difference and the reason S10.7 contains the largest numbers of avalanche
years in this analysis.
The range of return intervals across all paths (2 – 22.5 years) is similar to those reported for 12 avalanche
paths across a smaller spatial extent in the Chic Choc Mountains of Quebec, Canada (2 – 22.8) (Germain et
al., 2009). Although the authors in that study used a different avalanche signal index, this still suggests
considerable variation in avalanche frequency across avalanche paths within a region.



The return intervals for LJC in the Swan sub-region were the greatest and this is likely due to wildfire activity
in this path in 2003. LJC was heavily burned, and this created a steep slope with few trees that was once
moderately to heavily forested. Substantial anchoring and snowfall interception likely created an avalanche
path without many large magnitude avalanches for decades since slope forestation plays a substantial role in
runout distance and avalanche frequency in forested areas (Teich et al., 2012). In addition, wildfires in 1910
burned a majority of the JFS sub-region as well and the higher frequency of avalanche years recorded between
1910 and 1940 in S10.7 suggests this may also be a contributor to the high frequency of avalanche events in
that location (Reardon et al., 2008).
Our results also suggest that return interval increases as path length increases, though the sample size for this
correlation analysis on individual paths is small (n=12). This is likely because only large magnitude
avalanches affect the far extent of the runout of the path. This differs from a group of avalanche paths in
Rogers Pass, British Columbia, Canada, where path length was not significantly correlated with avalanche
frequency (Smith and McClung, 1997). However, that study used all observed avalanches, including artillery-
initiated avalanches, as opposed to a tree-ring reconstructed dataset.
JGO contains the maximum return interval for any path in the study, and the return intervals are significantly
different than numerous other paths. A lack of recording data after one large avalanche event could easily
skew this value. Another potential explanation is that this path is the only one located east of the Continental
Divide where the snowpack is often much shallower, particularly at lower elevations (Selkowitz et al., 2002),
thus inhibiting frequent large magnitude events from impacting the sampled runout zone. The fetch upwind
of this avalanche path is characterized by steep, rocky terrain harboring scoured slopes. This limits the
amount of snow available for transport to the JGO starting zone which may also influence the load and stress
placed upon this starting zone and subsequent large magnitude avalanches.
The greatest number of identified avalanche years is in the JFS sub-region. The avalanche paths in this sub-
region are all south or southeasterly facing whereas the other sub-regions span a greater range of aspects.
This may cause a bias toward a more unified representation of that aspect compared to the inclusion of other
aspects in the JFS sub-region.
The differences between sub-regions are likely due to localized terrain and weather factors and the interaction
of the two (Chesley-Preston, 2010). For example, Birkeland (2001) demonstrated significant variability of
slope stability across a small mountain range dependent upon terrain and weather. Slope stability and
subsequent large magnitude avalanching are likely to be highly heterogeneous across not only the sub-region,
but across a large region. This is also consistent with findings by Schweizer et al. (2003) that suggest
substantial differences in stability between sub-regions despite the presence of widespread weak layers.
**4.3 Regional Chronologies and RAAI**
The regional chronology we developed through the use of tree-ring analysis on collections made across 12
avalanche paths suggests, unsurprisingly, that the inclusion of more avalanche paths across a large spatial
extent produces a more robust identification of major avalanche winters. When we aggregate all 12 paths





together and apply established thresholds to discriminate the signal from the noise, we identified 30 avalanche
years throughout the region. This allows us to place each year in context of the region, or full extent of the
scale triplet, rather than simply collating all of the major avalanche winters identified in each individual path
or sub-region. However, we also account for the support and spacing by including adjacent avalanche paths
within a sub-region and multiple sub-regions throughout the region. This sampling strategy combined with
the large sample size collected throughout the region allowed for a robust assessment of regional avalanche
chronology derived from tree-ring records.
We tested the sensitivity of the term regional by removing specific and random paths. Our results suggest
that removing paths from this structure, and subsequently compromising the sampling strategy, introduces
noise. By reducing the sample size, we reduce the ability of the thresholds to filter out noise, thereby
increasing the actual number of avalanche years in the region. However, the sample size reduction also
reduces the number of identified avalanche winters common to the full 12 path regional record (Table 7).
Our results emphasize the importance of sampling more paths spread throughout the region of interest as well
as a large dataset across the spatial extent.
Avalanche path selection is clearly important when trying to assess avalanche frequency (de Bouchard
d'Aubeterre et al., 2019), and this is supported by our results suggesting that S10.7 is more influential than
any other path in our study (Table 7). However, selecting multiple paths throughout the region representing
a wide range of geomorphic characteristics and potentially influenced by local weather patterns provides a
reasonable assessment of regional avalanche activity in areas without historical records. By quantifying the
sensitivity of the number of avalanche paths within a given region, we illustrate that sampling a greater
number of avalanche paths dramatically increases the probability of identifying more avalanche years as well
as increases the ability to reconstruct major avalanche cycle chronologies. However, as previously noted,
dendrochronological techniques tend to underestimate avalanche frequency, which implies that caution
should be used when interpreting a regional avalanche signal using this technique, particularly as sample
numbers and qualities (e.g. cores vs. cross sections) decline.
Interestingly, the difference in median return interval throughout the "region" using 12 paths compared to
using four or eight paths changes only slightly. This suggests that fewer paths are still able to represent the
major avalanche return intervals across a region. However, choosing fewer paths appears to introduce more
noise and therefore fewer years identified than a regional chronology with more avalanche paths.
The RAAI provides a measure of avalanche activity scaled to the number of active avalanche paths across
the region through time. The years with the greatest RAAI value coincide with substantial activity provided
in the historical record as well as previous dendrochronological studies from the JFS sub-region (Butler and
Malanson, 1985a, b; Reardon et al., 2008). The winter of 1932-33 was characterized by heavy snowfall and
persistent cold temperatures leading to extensive avalanche activity that destroyed roadway infrastructure in
the JFS sub-region, 1950 saw a nearly month-long closure of U.S. Highway 2 due to avalanche activity, and
in 2002, an avalanche caused a train derailment. While these are all confined to the JFS sub-region, with the
exception of 2002, they are also years shared by at least two other sub-regions.


We examined the probability of detecting an avalanche year throughout the region by sampling any one given
path. In seven of 30 years, the $POD_{year}$ is only 8% and in all but three years the $POD_{year}$ is less than 40%. The
low $POD$ values are distributed throughout the time series, suggesting decreasing sample size back in time
or the number of active avalanche paths is not an influential factor. The POD is likely reflective of the spatial
variability of large magnitude avalanche occurrence across a region. It also aligns with the observational
findings of Mears (1992). During a major storm in 1986 throughout much of the western United States that
deposited 30-60 cm of snow water equivalent, Mears (1992) reports that in the area around Gothic, Colorado
less than 40% of avalanche paths produced avalanches and less than 10% produced avalanches approaching
the 100-year return interval. This also confirms the wide variability of avalanche years between sub-regions
recorded in our tree-ring record. Additionally, some of the avalanches in a given cycle may not be large
enough to be reflected in the tree ring record. Therefore, low values of $POD_{path}$ when considering only one
avalanche path and identifying only one common year of large magnitude avalanche activity (1982) amongst
the sub-regions through dendrochronology is not surprising. Paths with at least one scar (i.e. without applying
thresholds) during avalanche years identified in the regional chronology exhibit a greater $POD_{path}$, but this
greater $POD_{path}$ comes at the expense of introducing more noise if we were to simply use one scar per path
to define an avalanche event.
Our results also suggest that our sampling design using scale triplet increases the probability of detecting
avalanche activity across an entire region. We note that we are only able to scale our probability calculations
to our dataset with a limited historical observational record. However, our results illustrate the importance of
sampling more paths if the goal is to reconstruct a regional chronology. In our dataset, the greatest value of
$POD_{path}$ is 40% suggesting that if by chance, we sampled this path we would have captured the regional
avalanche activity 40% of the time.
The trends in RAAI over the entire period of record are likely influenced by the decreasing number of samples
available to record an event further back in time. Despite the RAAI accounting for the number of avalanche
paths (minimum of n = 3), the small sample size from the late 19th century precludes us from suggesting there
is a true increase in regional avalanche activity from 1867 to 2017. This is also supported by the absence of
positive or negative trend from 1950 to 2017 and 1990 to 2017.
**4.4 Limitations**
Overall, our results strongly suggest that sampling one path, or multiple paths in one sub-region, is
insufficient to extrapolate avalanche activity beyond those paths. Multiple paths nested within sub-regions
are necessary to glean information regarding avalanche activity throughout those sub-regions as well as the
overall region. Our study is still limited by the underrepresentation inherent in dendrochronological
techniques for identifying all avalanche events. While we analyzed 673 samples over the extent of the region,
some of the paths in our study had relatively small sample sizes per individual path as compared to recent
suggestions (Corona et al., 2012). This may have influenced the number of avalanche years identified and
subsequent return intervals per individual path. However, we attempted to limit the influence of sample size





by using full cross-sections from trees, robust and critical identification of signals in the tree-rings, and
appropriate established threshold techniques.
We also recognize that sampling more avalanche paths in our region would certainly provide a more robust
regional avalanche chronology, but time, cost, and resource constraints required an optimized strategy.
Finally, our study would undoubtedly have benefited from a longer and more accurate historical record for
comparisons and verification of the tree-ring record in all of the sub-regions. Overall, our study illustrates
the importance of considering spatial scale and extent when designing, and making inferences from, regional
avalanche studies using tree-ring records.
**5. Conclusions**
We developed a large magnitude avalanche chronology using dendrochronological techniques for a region
in the northern U.S. Rocky Mountains. Implementing a strategic sampling design allowed us to examine
avalanche activity through time in single avalanche paths, four sub-regions, and throughout the region. By
analyzing 673 samples from 12 avalanche paths, we identified 30 years with large magnitude events across
the region and a median return interval of ~3 years. Large magnitude avalanche return interval and number
of avalanche years vary throughout the sub-regions, suggesting the importance of local terrain and weather
factors. Our work emphasizes the importance of sample size, scale, and spatial extent when attempting to
derive a regional large magnitude avalanche chronology from tree-ring records. In our dataset, the greatest
value of $POD_{path}$ is 40% suggesting that if we sampled only this path, we would have captured the regional
avalanche activity 40% of the time. This clearly demonstrates that a single path cannot provide a reliable
regional avalanche chronology. Specifically, our results emphasize the importance of 1) sampling more paths
spread throughout the region of interest; 2) collecting a large number of cross-sections relative to cores; and,
3) generating a large dataset that scales to the appropriate spatial extent. Future work should include
conducting a similar study with a number of paths in the same sub-regions for verification, or in an area with
a more robust regional historical record for verification.





**6. Appendix A**
**Table A1: Regional chronologies from the International Tree-Ring Database used for cross-dating.**

| MT Avalanche Project Site | ITRDB Tree-Ring Chron. | Originator | Date Range | Species | Coordinates | Elevation | NOAA data set ID |
|---|---|---|---|---|---|---|---|
| Going-to-the-Sun Road sites | Going to the Sun Road (GTS) | Gregory T. Pederson Jeremy S. Littell | 1337 - 2002 | PSME | 48.42 -113.5167 | 1860M | noaa-tree-27540_MT159 |
| John F. Stevens Canyon sites | Doody Mountain (DOO) | Gregory T. Pederson Blase Reardon | 1660 - 2001 | PSME | 48.3833 -113.6167 | 1890M | noaa-tree-27536_MT155 |
| Lost Johnny Creek sites | Preston Park (PP) | Bekker, M.F.; Tikalsky, B.P.; Fagre, D.B.; Bills, S.D. | 1766 - 2006 | ABLA | 48.43 -113.39 | 2150M | noaa-tree-5993_MT117 |
| Red Meadow sites | Numa Ridge Falls (NRF) | Gregory T. Pederson Brian Peters | 1645 - 2001 | PSME | 48.51 -114.12 | 1695M | noaa-tree-27550_MT168 |

**Table A2: Avalanche Years identified in the regional analysis (Region, n=29) and avalanche years identified in**
**one or more paths in the individual avalanche path analysis (Ind. Paths Unique Years, n=49). Years in bold**
**indicate years in common between the two sets (n=27).**

| Region | Ind. Paths Unique Years |
|---|---|
| **1866** | **1866** |
| **1872** | **1872** |
| **1880** | **1880** |
| | 1907 |
| | 1912 |
| | 1913 |
| | 1923 |
| **1933** | **1933** |
| **1936** | **1936** |
| | 1943 |
| 1945 | |
| **1948** | **1948** |
| | 1949 |
| **1950** | **1950** |
| **1954** | **1954** |
| 1956 | |





| | |
|---|---|
| **1965** | **1965** |
| | 1966 |
| | 1967 |
| | 1968 |
| **1970** | **1970** |
| **1971** | **1971** |
| **1972** | **1972** |
| **1974** | **1974** |
| **1976** | **1976** |
| | 1979 |
| **1982** | **1982** |
| | 1983 |
| | 1985 |
| | 1986 |
| | 1987 |
| | 1989 |
| **1990** | **1990** |
| | 1991 |
| | 1992 |
| **1993** | **1993** |
| | 1995 |
| | 1996 |
| **1997** | **1997** |
| **1998** | **1998** |
| | 1999 |
| | 2001 |
| **2002** | **2002** |
| **2003** | **2003** |
| **2004** | **2004** |
| **2009** | **2009** |
| | 2010 |
| **2011** | **2011** |
| **2012** | **2012** |
| **2014** | **2014** |
| **2017** | **2017** |


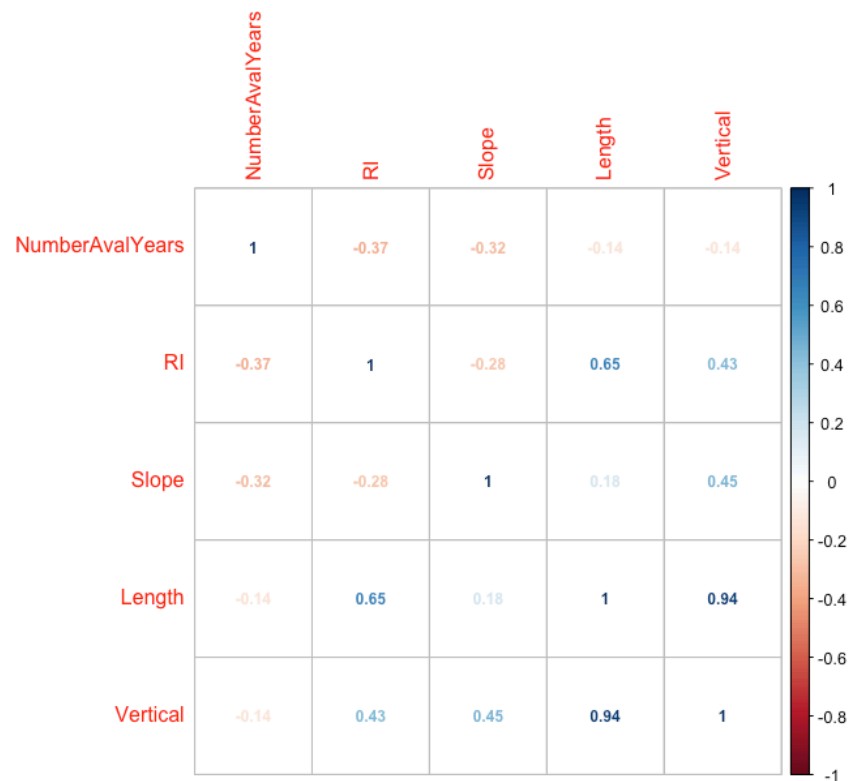

**Figure A1: Correlation matrix (Pearson correlations coefficients) of the number of avalanche years, return interval (RI), starting zone slope angle (Slope), and path length (Length).**

## 7. Data availability

Data for this work can be found in ScienceBase repository: Peitzsch, E. H., Stahle, D. K., Fagre, D. B., Clark, A. M., Pederson, G. T., Hendrikx, J., and Birkeland, K. W.: Tree ring dataset for a regional avalanche chronology in northwest Montana, 1636-2017. U.S. Geological Survey., U.S. Geological Survey data release, https://doi.org/10.5066/P9TLHZAI, 2019.

## 8. Author contribution

EP responsible for study conception and design, data collection, analysis, writing. JH contributed to development of study design, methods, editing, and writing. DS responsible for data collection, tree-ring processing and analysis and writing. GP, KB, and DF contributed to study design, editing and writing.



**9. Disclaimer**
Any use of trade, firm, or product names is for descriptive purposes only and does not imply endorsement by
the U.S. Government.
**10. Acknowledgements**
We extend gratitude to Adam Clark for his substantial data collection efforts and Zach Miller for his
assistance processing samples. This work was supported by the USGS Land Resources Western Mountain
Initiative project.

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
