# Peer review of "A regional spatio-temporal analysis of large magnitude 2 snow avalanches using tree rings"

_Natural Hazards and Earth System Sciences, 2020_

## Referee Comment (RC1) · Adrien Favillier (Referee) · 14 Sep 2020

GENERAL COMMENTS

In this article, the authors aim to reconstruct the snow avalanche activity over the two last centuries and at a regional scale. They used dendrogeomorphic technics and its last methodological developments to reach this goal. Through the careful analysis of 647 trees within 12 studied avalanche paths (one of the widest sampling in the discipline), the authors successfully reconstructed 49 events between 1866 and 2017 and thus derive recurrence interval at relevant scales both for hazard planning and climate change related research. In addition to the newest data provided for this part of

the world (last ones were published in 1979-85, at the "regional scale" and 2008 at the path scale), the authors propose relevant questions and solutions on how to combine snow avalanche chronologies derived from tree-rings at path scale to a regional snow avalanche activity, with specific focuses about the sampling strategy and the expansion to lower spatial scale. The availability of the data is appreciated.

From a global point of view, this article is well written and matches international standards. This work, its results and their discussion deserve to be published in NHESS. However, some corrections and clarifications preclude from a publication of this paper.

First of all, as a non-native speaker, some paragraphs, especially related to the numerous abbreviations (although necessary), remain complex and sometimes not fluent enough to clearly understand the developed idea at the first reading. Revising the manuscript with the aim to make it clearer and easy/fluent to read would be great, especially for non-native speakers.

Second, there is a major mistake in the Wit formula (2) which partially distorts the results. At the difference of Kogelnig-Mayer et al. (2011), in the 4-Steps procedure developed by Favillier et al. (2017, 2018), the weighted sum of the first term is not multiplied by the total of growth disturbance of the year t. To use the Wit threshold initially defined in Favillier et al. (2017, 2018), please use the formula presented at the pages 93 and 14, respectively, of these articles. Otherwise, please define new thresholds that could represent your range of values. At the end, results should be nearly the same as you had the opportunities to work with many cross-sections.

SPECIFIC COMMENTS Finally, several additional remarks are listed below:

L.60-66: According to the table title, Table 1 appears incomplete (21 references over the 42 existing studies with more than one avalanche path). Precisely, what were your selection criteria? Please either clarify the caption or add the missing references.

L.124-126 (Table 2): Please, modify "n" for "Trees (n)" or "Nb. of trees (n)" in order to

be clearer that "n" is for the number of sampled trees per path.

L.167: As you mainly worked with dead trees, how did you deal with? Did you take account of their year of death? Did you take account of the forest age structure of the path to suspect past high magnitude event that partially destroyed the forest?

L.197-204: This comparison makes sense in a general methodological point of view (how much growth disturbance are we missing using core instead of cross-section). However, it does not match the main aims of the article and, accordingly, complexifie the reading. I suggest removing the comparison and the related paragraphs, but to discuss the advantages/limits to work with cross-sections in the Discussion section. On the one hand, knowing all the growth disturbances strengthen the reliability of your reconstructions. On the other hand, cross-sections are usually taken out from dead trees, so you cannot really assess the age structure of the in-place forest. Moreover, it is time consuming to process in comparison to cores, as you will have to carefully analyze the whole section. Lastly, it is an exceptional situation, as in Europe, we are mostly working on living trees in protection forest.

L.299 (Fig. 3): In my opinion, the term "event" is not really suitable as it refers to cross-dated growth disturbances and not to a reconstructed avalanche event. Responses, as mentioned in the figure title, could fit.

L.327-328 (Fig. 5): The name of the Y-axis and the captions are not really clear. I suggest replacing "Avalanche path" by "Avalanche event."

L.380 (Fig. 8): Graphs (a) and (c) could deserve a secondary axis for (a) the sample size and (c) the number of avalanche path. It would be easier to read. Here is the R-code I use to plot a secondary axis:

ylim.prim <- c(0, max(Growth Disturbances, na.rm = T)) # Primary axis: distance to zero ylim.sec <- c(0, max(Sample Size, na.rm = T)) #Secondary axis: distance to zero b <- diff(ylim.prim)/diff(ylim.sec) #Computing multiplicative coefficient a <- b*(ylim.prim[1]

- ylim.sec[1]) #Distance to zero

ggplot(data, aes(x=Years))+ geom_line(aes(y = Growth Disturbances))+ #Primary axis geom_line(mapping = aes(y = a+Sample Size*b))+ #Secondary axis scale_y_continuous(sec.axis = sec_axis($\sim$ (. - a)/b, name = "Sample Size"))

L.432–433: What is the purpose of this comparison? I suggest removing it to simplify the manuscript.

TECHNICAL COMMENTS Please, carefully revise the manuscript to tackle the typos. Most of them are located in the figure references in the text (extra brackets).

---

## Author Comment (AC1) · 25 Sep 2020

We appreciate the thorough review of our manuscript and the constructive feedback provided by Adrien Favillier. Here, we address each comment and reference the changes in the revised manuscript. We are pleased that the reviewer recommends this work be published in NHESS after the appropriate revisions are addressed. We've posted the responses below for timely discussion and will provide a complete revised manuscript after receiving comments from other reviewers and the editor.

*Comment*:
First of all, as a non-native speaker, some paragraphs, especially related to the numerous abbreviations (although necessary), remain complex and sometimes not fluent enough to clearly understand the developed idea at the first reading. Revising the manuscript with the aim to make it clearer and easy/fluent to read would be great, especially for non-native speakers.
*Response*:
We revised the writing to be more succinct and accessible throughout the manuscript.

*Comment*:
Second, there is a major mistake in the Wit formula (2) which partially distorts the results. At the difference of Kogelnig-Mayer et al. (2011), in the 4-Steps procedure developed by Favillier et al. (2017, 2018), the weighted sum of the first term is not multiplied by the total of growth disturbance of the year t. To use the Wit threshold initially defined in Favillier et al. (2017, 2018), please use the formula presented at the pages 93 and 14, respectively, of these articles. Otherwise, please define new thresholds that could represent your range of values. At the end, results should be nearly the same as you had the opportunities to work with many cross-sections.
*Response*:
Thanks for the careful attention to detail on this equation. Fortunately, this was simply a typo in the equation in the manuscript. We used the eqn. from Favillier et al. (2017, 2018) in the analysis, but accidentally inserted the $R_t$ term from the Kogelnig-Mayer et al. (2011) eqn. in the manuscript. Equation (2) revised to reflect this.

*Comment*:
L.60-66: According to the table title, Table 1 appears incomplete (21 references over the 42 existing studies with more than one avalanche path). Precisely, what were your selection criteria? Please either clarify the caption or add the missing references.
*Response*:
We included only the initial studies using a dataset with more than one avalanche path. In other words, if a study used the same dataset again in subsequent work, then, instead of repeating this in the table, we omitted it. We revised the table caption to clarify this.

*Comment*:
L.124-126 (Table 2): Please, modify "n" for "Trees (n)" or "Nb. of trees (n)" in order to be clearer that "n" is for the number of sampled trees per path.
*Response*:
2nd column in Table 2 changed to "Trees (n)".

*Comment*:
L.167: As you mainly worked with dead trees, how did you deal with? Did you take account of their year of death? Did you take account of the forest age structure of the path to suspect past high magnitude event that partially destroyed the forest?
*Response*:
Yes, we took into account the death year. We did not quantify or relate any death dates to avalanches that fell outside of observed avalanche mortality events. We assigned C-1 events to trees that were known to

be killed by an avalanche impact, in place trees with earlywood growth for the year of the observed avalanche event.

*Comment*:
L.197-204: This comparison makes sense in a general methodological point of view (how much growth disturbance are we missing using core instead of cross-section).
However, it does not match the main aims of the article and, accordingly, complexifie the reading. I suggest removing the comparison and the related paragraphs, but to discuss the advantages/limits to work with cross-sections in the Discussion section. On the one hand, knowing all the growth disturbances strengthen the reliability of your reconstructions. On the other hand, cross-sections are usually taken out from dead trees, so you cannot really assess the age structure of the in-place forest. Moreover, it is time consuming to process in comparison to cores, as you will have to carefully analyze the whole section. Lastly, it is an exceptional situation, as in Europe, we are mostly working on living trees in protection forest.
*Response*:
We believe that the results of this analysis in examining cores vs. cross-sections align with the objectives of the article. Determining the value of using cross-sections was central to the original sampling design which distinguishes this study from previous dendro-avalanche research. The exercise must be mentioned in the Methods section because we present values on the comparisons in the Results and then the Discussion. Lastly, we provide this comparison to simply quantitatively illustrate the difference in using cross sections vs. cores and do not in any way discount any studies that use cores.
Regarding the "age structure of the in-place forest": we did not sample/process GDs to identify stand replacing large magnitude avalanche events using age structure cohort recruitment methods, similar to those used in fire or other ecological studies. The stems sampled from mixed subalpine forests were of great variation in ages. All the forests we worked in across the northern Rockies have mixed age structure due to varied disturbances, so quantifying a stand age structure is usually not possible, or all that informative relative to the growth disturbance information provided by cross-sections. Appropriately, we offer no unsubstantiated conjecture relating previous large magnitude events to the age structure of trees sampled in the paths. While it can be more time consuming to process cross-sections vs. cores, we utilized the opportunity to collect cross-sections for a more complete perspective and ability to identify GDs with greater confidence.
Regarding the last sentence in the comment: we were also constrained from sampling trees within a "protected forest" as a U.S. National Park prohibits wood collection on its lands except in the case of permitted research.

*Comment*:
L.299 (Fig. 3): In my opinion, the term "event" is not really suitable as it refers to crossdated growth disturbances and not to a reconstructed avalanche event. Responses, as mentioned in the figure title, could fit.
*Response*:
X-axis title for Panels (a) and (b) changed to "Classification of Each Response" to fit with the Y-axis title.

*Comment*:
L.327-328 (Fig. 5): The name of the Y-axis and the captions are not really clear. I suggest replacing "Avalanche path" by "Avalanche event."
*Response*:
The Y-axis title should read "...Avalanche Paths" as it currently stands. This figure highlights the number of avalanche paths in which an avalanche event occurred in any given year. We revised the figure caption to be clearer.

*Comment*:

L.380 (Fig. 8): Graphs (a) and (c) could deserve a secondary axis for (a) the sample size and (c) the number of avalanche path. It would be easier to read. Here is the R-code I use to plot a secondary axis: ylim.prim <- c(0, max(Growth Disturbances, na.rm = T)) # Primary axis: distance to zero ylim.sec <- c(0, max(Sample Size, na.rm = T)) #Secondary axis: distance to zero b <- diff(ylim.prim)/diff(ylim.sec) #Computing multiplicative coefficient a <- b*(ylim.prim[1] - ylim.sec[1]) #Distance to zero ggplot(data, aes(x=Years))+ geom_line(aes(y = Growth Disturbances))+ #Primary axis geom_line(mapping = aes(y = a+Sample Size*b))+ #Secondary axis scale_y_continuous(sec.axis = sec_axis(_ (. - a)/b, name = "Sample Size"))

*Response*:

First, thank you very much for the R-code. We greatly appreciate it. We contemplated, at length, adding a secondary axis for these exact plots prior to submission. We typically use one primary axis for ease of interpretation. However, in this case we feel the secondary axis might be appropriate. As such we revised the figure explicitly note the secondary axis for panels (a) and (c) in the new caption for Figure 8.

[Figure]

*Comment*:
L.432–433: What is the purpose of this comparison? I suggest removing it to simplify the manuscript.
*Response*:
We removed this sentence as it did not fit within the context of this paragraph.

*Comment*:
TECHNICAL COMMENTS Please, carefully revise the manuscript to tackle the typos. Most of them are located in the figure references in the text (extra brackets).
*Response*:
We corrected the typos throughout the manuscript.

---

## Referee Comment (RC2) · Brian Luckman (Referee) · 29 Sep 2020

Brian Luckman (referee) Luckman@uwo.ca September 22nd, 2020

GENERAL COMMENTS This paper presents snow avalanche histories from 12 avalanche tracks, 3 from each of four regions in the Northern Rocky Mountains of N.W Montana based on tree-ring data from 637 trees. These data are used to define the history and frequency of large magnitude avalanches for individual tracks, sub-regions (mountain ranges) and across a region of ca 3000 km 2. The paper then estimates the

efficiency of using various combinations of these chronologies to estimate a regional chronology of high magnitude avalanches in order to guide future sampling strategies for estimating regional avalanche activity. The techniques used are based on prior usage from the literature and, to my knowledge, the attempt to assess the efficiency of developing a regional history is novel.

The paper is well written but overlong and I have many questions of detail. The use of symbols to identify important terms is difficult to follow e.g Wit , RAAIt etc and a table describing these terms (in words) would be a useful addition. One of the principal difficulties is the comparison of statistics such as RI values between sites based on records of different length where the RI values are strongly related to survival of older individuals within the avalanche path. Perhaps a comparison based on e.g. the last fifty years would be better to compare differences between tracks.

No indication is given of the number of living vs dead trees sampled. If one discounts the first 10 years of record over a third of the trees sampled have <35 years and half <60 years of record. How large/ tall are these trees on average at these ages and how might the nature of the tree-ring signal (i.e. the probability of recording a given event) vary with the age /height/ robustness of the tree. The avalanche chronologies are strongly biased towards the lifetime and response characteristics of the trees sampled. Although the number of GDs is cited in several places the breakdown of the individual types of GD e.g. scars, reaction wood series, TRD, tree mortality, etc. is never given.

In several cases the results are self evident- one gets better results from more sites, more trees, cross sections vs cores. The main strength is the regional and sampling approach. However, I have reservations about some of the derived statistics and the comparisons between individual records. There is little comment on the variability of the records within each of the sampling regions, or for example, the similarity between two adjacent paths. The main focus is the regional comparison. This regional approach tacitly assumes no significant differences in avalanche climate, or triggering factors across the region. There is no specific exploration of the relationship between

avalanche activity and climatic factors. I think this paper needs revision to address some of the concerns addressed below

DETAILED and SPECIFIC comments Line Comment

52 delete semi-colon before bracket 66 Most of the data in table 2 is not greatly relevant to the paper. It is simply a compendium of earlier chronology studies. It is not used and could be in an appendix or supplementary material. 85-8 In this paper large magnitude avalanches are identified based on the cumulative evidence of disturbance by avalanches for an individual year in a given track given a minimum number of trees sampled. This identification is independent of the location of these disturbances within the individual avalanche track. The distribution of sampled trees within the avalanche track is therefore critical to the interpretation of this evidence with respect to avalanche hazard. Large magnitude in this scenario doesn't necessarily mean large or full length avalanches that would impact the runout/ danger zones. The authors need to emphasise more strongly that these avalanche chronologies are based on sampling in the terminal zone and down track margins and therefore the large magnitude events are inferred to be large full length avalanches that would represent hazard to these areas. In some cases there are a significant number of samples in the upper part of the track. 118 Figure 1 When enlarged Figure 1 clearly shows black dots which are assumed to be the sampling locations within the tracks. The figure caption should clearly point this out-it is not clear from the key and does not seem to be mentioned in the text or caption. As tree location is a critical factor in defining the size of avalanches their specific location is important. At several tracks the location of the sampled trees is some distance from the terminal zone of the avalanche track (there are two sampled areas on the northernmost GTSR site near Crystal Point?). Perhaps the track names should be identified on Figure 1. Tables 2, 4 and 5. Is the colouring necessary? 110 These data do not appear to be used or referenced in the present study, even for comparative purposes. Did they identify similar major avalanche events? 160 the spatial footprint is 3000km2 in the abstract and 3500 km2 here 167 The text at this point suggests that

all the cross sections were from dead trees and that the only living trees sampled were cored. Is this the case? Is the outer ring from these dead trees assumed to be from a "high magnitude" event i.e. the tree was killed/ sheared by an avalanche. These outer rings were presumably crossdated from adjacent living trees or chronologies. Were the core data actually used? 200 + How does one also counter the censoring of the avalanche record due to continuing persistence of damage (e.g. reaction wood or TRD) in tree rings for several years following a major disturbance? 212 responses within the tree or over the site? 212-4 should the analyses and comparison of return intervals be limited to a common period when there is a reasonable sample of avalanche events (however defined) based on the age distribution of sampled trees within all tracks? 221 Figure 2 More information needs to be given in the caption identifying the symbols used N= sample trees available. GD= number of GDs identified. Perhaps include (N) after sample size in line 226. Is GD any GD or those above some minimal value? The context seems to indicate it is the number of GDs identified and not their magnitude. 229 Is the statistic for avalanche years simply binary i.e. yes no? 239-40 therefore high magnitude years are all years where Wit is $\geq$3? Is the last term in Eqn 2 simply It? Essentially you derive a Wit value for each year for which there was avalanche data in each track and identify avalanche years as those with Wit $\geq$ 0.3. Line 229 in the text indicates that RI calculations are based on the avalanche year examples (box 2 of fig 2) but lines 241 et seq. indicate that RI values are also calculated for high magnitude events (Wit $\geq$ 0.3) only. Therefore are there two sets of RI data for (i) avalanche years and (ii) high magnitude (Wit) years? So which data are used in the subsequent analyses? Are these high magnitude years simply binary data (yes/no?) 253 RAAIt is based on the definition of avalanche years (It), not high magnitude Wit years. Therefore avalanche years are identified using the It statistic but high magnitude avalanches are identified using the Wit statistic. It appears that the RI data are calculated based on both the It and the Wit classification whereas the RAAIt statistics are based on the It definition of avalanche years. Is this correct? Line 324 seems to imply that the avalanche years identified in Figure 2 and the high magnitude events identified

using Wit were identical so this difference does not matter? In any event only one set of calculations defining RI values should be specified. The term RI is used throughout the text but in places it is not clear whether it refers to the mean or median value. 263 how does the probability of detection differ from the probability of avalanches? 284 are these comparisons included in this paper? 291 ID by GD class but not type? So what was principal evidence used? 298 this is predictable given the ages of trees sampled. Perhaps more interesting would be the years with the highest It values 299 Figures 3a and b appear identical and one is redundant. The scale on Fig 3b is incorrect (0.3 %?). The ages in Fig 3c indicate that many of these trees were quite small. What would the diameter of a 40 year old tree be? How does age influence the nature of the GD? In 3d were the larix and betula species identified? 308 missed 67 or 66? 312 Figure 4 needs a scale. Some comparison of the derived GD data would be useful to make the point. To be effective this topic warrants a more extensive discussion and presentation of data than that presented here. This discussion and figure should probably be deleted. 312 More importantly were results from these cores actually used in the analysis. 324 Table 5 what does the standard deviation figure refer to (bottom line). Explain in caption? 332 Tables 4 and 5 What is the statistic 1/RI in these tables? Why is the median RI value used rather than the mean? Explain in the caption 334 Table 4 An additional line identifying the sub region should be added to the top of the table. The table should also give the period of record utilised to calculate RI for each track. 337 JGO is a function of the early record but why LJB? LJB and LJC are 26, LGA is 25 and shed 7 is 28? 399 Table 7 explain MLC and HLC in the caption 346 Figure 6 what are the data plotted in this Figure? The median of GDO in Table 4 is ca 34 but in this figure it is ca 28. For LGP the median is ca 12.6 in Table 4 but ca 8 in figure 6 338-40 surely the similarities and differences between tracks reflect the length and nature of the avalanche record in each track? Differences/ similarity in return intervals are partially dependant on the length of record 349-66 These differences in recurrence intervals are calculated for different periods of record. To be comparable do they need to be calculated over the same interval? 369 The avalanche records in these tracks start in 1933, 1936 and 1993

so why compare them to a record starting in 1908 which presumably has avalanches predating those records. Surely comparisons need to be over the same intervals? 370 Figure 7 a nice ( original?) way to show these data 380 Figure 8b the red line is not visible. Perhaps delete it and simply indicate this value in the caption. 386 et seq. But the records being compared have quite different lengths and histories. How unique is the record of individual paths? If you compared the record of the tracks with similar length of record (say, ca 1950-2010, RMA-C, 54.3, LJA, 10.7 and S4.7) how similar are they? 405-6 These trends are mainly an effect of the increased sampling of avalanche years 432-3 relevance of these comparisons? 435 up into the bottom? English? Is the bottom the end or center of the track? 437-40 some specific dates needed here as this is the basis for the selection of records used. What specifically is the most recent time period for which you have adequate data across the network? 451-2 how frequent is tree removal? What % of GDs are termination of growth vs other indicators of avalanche damage? 454 Although mentioned several times this incomplete historical record is never presented or directly compared to the equivalent tree-ring record for the comparable sites. 475 the difference between Readon's earlier results or the other avalanche tracks? What are these differences? 481 LJC has the greatest RI? It has the greatest median but not mean. The large median is a function of the small sample size in this track. The fire may have taken out evidence for most events between 1943 and 2017 and therefore this is not a valid comparison. 489 using which RI value, mean or median? What is the correlation statistic? 495 JGO is very unusual with only two avalanches between 1880 and 2017! The critical difference is the absence of documented events in the last 50 years. The only answer to these tentative explanations is more data from adjacent tracks. Perhaps the only comment necessary is that the reason for this is not known 507 but these differences are never explored. 527 but these changes are also influenced by which avalanche track you remove. 532 But how typical is the record of s10-7 of other paths in the region see e.g. Table 4 533 s10.7 has the most avalanche activity but surprisingly is not compared with the available, if limited, observational record. 538 What is an avalanche cycle chronology? 547-9 these data

[Figure]

would be useful here to validate some of these comments or are they solely based on the examples which follow. 566 paths with one scar or one GD of class 3? Where are these scar data? 570 is the sample design or the number of paths the critical factor here? The sample design clearly increases the area covered. 582-5 Basic point is that if you sample more avalanche tracks you get more avalanche years and a more consistent pattern may emerge. However the pattern of avalanche activity varies from track to track and from year to year 587 Is this a function of sample size or other characteristics such as the time period covered by those samples and the sampling network? 603 this median value probably should be linked to a time frame to which it applies

Figure A1 What are the data used here (reference to Table 2)? Some numbers are barely visible. Perhaps use bolder (larger) numbers and colour as a background to individual cells?

---

## Referee Comment (RC3) · Brian Luckman (Referee) · 29 Sep 2020

Dear Authors,

The review was originally submitted on Sept 23 but there were difficulties with communications hence the delay. The comments are keyed to the original manuscript and I had not seen the response to referee 1. Unfortunately the formatting of my detailed comments was lost during the upload. I apologize for this

Yours sincerely Brian Luckman
* * *
2020-253, 2020.

---

## Author Comment (AC2) · 24 Oct 2020

We appreciate the thorough review of our manuscript and the constructive feedback provided by Brian Luckman. Here, we address each comment and reference the changes in the revised manuscript.

*Comment:*
The use of symbols to identify important terms is difficult to follow e.g Wit, RAAIt etc and a table describing these terms (in words) would be a useful addition. One of the principal difficulties is the comparison of statistics such as RI values between sites based on records of different length where the RI values are strongly related to survival of older individuals within the avalanche path. Perhaps a comparison based on e.g. the last fifty years would be better to compare differences between tracks.
*Response:*
We incorporated verbal descriptors of these terms in Figure 2 so as to not increase the length of the manuscript.

*Comment:*
One of the principal difficulties is the comparison of statistics such as RI values between sites based on records of different length where the RI values are strongly related to survival of older individuals within the avalanche path. Perhaps a comparison based on e.g. the last fifty years would be better to compare differences between tracks.
*Response:*
We subset the period of record for each path from 1967-2017 and compared RI values. Nine paths exhibit no change in RI values when compared to the full record and one path RI values decreased by 4 years. We observed larger changes in the other two paths; JGO path where only one avalanche year was recorded (down from 5) and the median RI in LJC changed from 22.5 years to 35 years. We previously discussed JGO and LJC and the variable RIs of each of those paths in the Discussion. This exercise highlights that discussion emphasizing that these two paths were indeed slightly different than the others. We added the above text (ca. 350-353) to illustrate that we examined the most recent 50 years to "scale" the return periods to account for loss of older trees.

*Comment:*
No indication is given of the number of living vs dead trees sampled. If one discounts. the first 10 years of record over a third of the trees sampled have <35 years and half <60 years of record. How large/ tall are these trees on average at these ages and how might the nature of the tree-ring signal (i.e. the probability of recording a given event) vary with the age /height/ robustness of the tree. The avalanche chronologies are strongly biased towards the lifetime and response characteristics of the trees sampled. Although the number of GDs is cited in several places the breakdown of the individual types of GD e.g. scars, reaction wood series, TRD, tree mortality, etc. is never given.
*Response:*
We added # of living vs. dead trees sampled (line 290); 539 dead and 116 live sampled. Given the regulations of the protected areas in which we sampled, the majority of our cross sections were dead trees. The only exception is if the tree was growing a new leader and we sampled the old top that was previously destroyed in an avalanche. In addition, the dead trees spanned the same age class structure as the living trees from the surrounding forest and in the runout zone. Therefore, we did not bias the record by only working with dead trees. Rather, we simply improved the overall quality of the data by being able to work with cross sections. We would likely have obtained only lower quality responses by sampling more cores. Further, there were very few live trees/samples within the avalanche path itself and we sampled those that do exist.

*Comment*:
In several cases the results are self evident- one gets better results from more sites, more trees, cross sections vs cores. The main strength is the regional and sampling approach. However, I have reservations about some of the derived statistics and the comparisons between individual records. There is little

comment on the variability of the records within each of the sampling regions, or for example, the similarity between two adjacent paths. The main focus is the regional comparison. This regional approach tacitly assumes no significant differences in avalanche climate, or triggering factors across the region. There is no specific exploration of the relationship between avalanche activity and climatic factors.
*Response:*
In the discussion we provide interpretation on the variability between paths (e.g. JGO located east of the Continental Divide, LJC burned in the past, S10-7 sampled slightly differently than others b/c it was from another study). We also added text (lines 522-531) explaining inherent variability of RIs between individual paths for a variety of reasons. For example, avalanches are a function of weather and snowpack structure/variability. Climate drives weather, but is not a first order effect on avalanche occurrence in any one given avalanche path. This is the motivation for this study. We derive a regional avalanche chronology to provide a spatial scale that aligns more with the spatial scale of climate drivers than any one individual path. On that note, an analysis of climate drivers of avalanche frequency is beyond the scope of this manuscript. Climate and regional avalanche relationships are the topic of a follow-on manuscript using this dataset that is currently undergoing peer-review.

Detailed and specific comments:
*Comment:*
52 delete semi-colon before bracket
*Response:*
Removed

*Comment:*
66 Most of the data in table 2 is not greatly relevant to the paper. It is simply a compendium of earlier chronology studies. It is not used and could be in an appendix or supplementary material.
*Response:*
This table places our study in context to other studies re: spatial extent, sample size, # of GDs, etc. However, in an effort to decrease the length of our manuscript we moved it to Appendix A.

*Comment:*
85-8 In this paper large magnitude avalanches are identified based on the cumulative evidence of disturbance by avalanches for an individual year in a given track given a minimum number of trees sampled. This identification is independent of the location of these disturbances within the individual avalanche track. The distribution of sampled trees within the avalanche track is therefore critical to the interpretation of this evidence with respect to avalanche hazard. Large magnitude in this scenario doesn't necessarily mean large or full length avalanches that would impact the runout/ danger zones. The authors need to emphasize more strongly that these avalanche chronologies are based on sampling in the terminal zone and down track margins and therefore the large magnitude events are inferred to be large full length avalanches that would represent hazard to these areas. In some cases there are a significant number of samples in the upper part of the track.
*Response:*
You are correct in that the definition of large magnitude avalanche in this study doesn't necessarily mean full length of avalanche path as we indeed sample at various locations in the runout zone and into the track in some instances. However, we sampled spatial extents within each avalanche path that represent large avalanches as defined in Greene et al. (2010). The areas sampled are representative of the runout extents of $\geq$ size D2 avalanches. We also used recent (within previous 10 years) observed large magnitude avalanche activity in these paths to constrain the spatial extent of our sampling. We added this text line 150.

*Comment*:
118 Figure 1 When enlarged Figure 1 clearly shows black dots which are assumed to be the sampling locations within the tracks. The figure caption should clearly point this out-it is not clear from the key and does not seem to be mentioned in the text or caption. As tree location is a critical factor in defining the size of avalanches their specific location is important. At several tracks the location of the sampled trees is some distance from the terminal zone of the avalanche track (there are two sampled areas on the northernmost GTSR site near Crystal Point?). Perhaps the track names should be identified on Figure 1.
*Response*:
We added more description in the figure caption in addition to the legend. As noted in the response above all sampling locations are within spatial extents representative of large magnitude avalanche extents.

*Comment*:
Tables 2, 4 and 5. Is the colouring necessary?
*Response*:
We removed the color.

*Comment*:
110 These data do not appear to be used or referenced in the present study, even for comparative purposes. Did they identify similar major avalanche events?
*Response*:
These studies are referenced in the Intro. and Discussion (ca lines 153, 484, 500, 568) and in current Table A1 for comparison of spatial extent, sample size, etc.

*Comment*:
160 the spatial footprint is 3000km2 in the abstract and 3500 km2 here.
*Response*:
Revised to read 3500 $km^2$ in the Introduction (ca. line 92).

*Comment*:
167 The text at this point suggests that all the cross sections were from dead trees and that the only living trees sampled were cored. Is this the case? Is the outer ring from these dead trees assumed to be from a "high magnitude" event i.e. the tree was killed/ sheared by an avalanche. These outer rings were presumably crossdated from adjacent living trees or chronologies. Were the core data actually used?
*Response*:
First, most cross sections were from dead trees. As previously mentioned, the only exception is if the tree was growing a new leader and we sampled the old top that was destroyed in an avalanche. The outer ring from these dead trees is not presumed to be from an avalanche as the tree may have died from some other cause and transported to the sampled location. We only dated the outer ring as an avalanche if historical records indicated a large magnitude avalanche in that path. The cross sections were indeed cross dated from living trees (i.e. cores) from either the adjacent gallery forest or nearby chronologies from the ITRDB (see Table A2). The core data were used for cross dating and for avalanche event dating if a signal was evident as some of these cores were sampled near the trim line where very large avalanches may have reached.

*Comment*:
200 + How does one also counter the censoring of the avalanche record due to continuing persistence of damage (e.g. reaction wood or TRD) in tree rings for several years following a major disturbance?
*Response*:
Cross sections provide us the ability to scrutinize reaction wood in any given location along the sample relative to other parts of the tree. Working with cross sections from all across the runout zone more or less ensures we don't miss avalanche events in years subsequent to a major slide event that damaged many

trees. Subsequent slide events not only become obvious in trees that have recorded a major event already due to the generation of new scars and reaction wood growth that forms in different cardinal directions due to impacts from differing predominant flow directions, but also some proportion of trees in different parts of the runout zone that were not damaged in the prior slide event are likely to have captured the new event that occurred a year or two after the major event that was identified and classified. We see this throughout the record where individual avalanche events can and do classify as major slide event occurring in the same path but only a year or two after a different major event.

In addition to carefully classifying each signal (GD) in each sample using the classification scheme (see current Table 2), we also made the best attempt possible to filter out the the noise by using recent threshold methods devised by Corona et al. (2012) and Favillier et al. (2018, 2018)/Kogelnig Mayer (2011). The classification scheme clearly delineates that reaction wood or TRD alone receives a lower ranked classification. This then is taken into account in the $W_{it}$ indexing process.

*Comment*:
212 responses within the tree or over the site?
*Response*:
The number of responses per year were calculated for each avalanche path. Descriptive statistics were computed for each path, sub-region, and region. We clarified the text to read: "We calculated the age of each tree sampled, the number of responses per year in each avalanche path, and computed descriptive statistics for the entire dataset." (lines 211-212)

*Comment*:
212-4 should the analyses and comparison of return intervals be limited to a common period when there is a reasonable sample of avalanche events (however defined) based on the age distribution of sampled trees within all tracks?
*Response*:
See response above to where we examined RIs from 1967-2017 as recommended with no major difference except in two paths. We previously discussed these paths in the Discussion.

*Comment*:
221 Figure 2 More information needs to be given in the caption identifying the symbols used N= sample trees available. GD= number of GDs identified. Perhaps include (N) after sample size in line 226. Is GD any GD or those above some minimal value? The context seems to indicate it is the number of GDs identified and not their magnitude.
*Response*:
We added text to the caption of Figure 2 and added "(N)" to line 227. GD is any growth disturbance identified and classified (as per Table 2) due to an avalanche.

*Comment*:
229 Is the statistic for avalanche years simply binary i.e. yes no?
*Response*:
Yes.

*Commment*:
239-40 therefore high magnitude years are all years where Wit is ≥3? Is the last term in Eqn 2 simply It? Essentially you derive a Wit value for each year for which there was avalanche data in each track and identify avalanche years as those with Wit ≥ 0.3.
*Response*:

We identify a large magnitude avalanche year as one where $W_{it}$ is ≥2 (a measure of Medium and High confidence). The last term in Eq. 2 is indeed a typo. As we mentioned to Reviewer 1: We used the eqn. from Favillier et al. (2017, 2018) in the analysis, but accidentally inserted the $R_t$ term from the Kogelnig-Mayer et al. (2011) eqn. in the manuscript.

*Comment*:
Line 229 in the text indicates that RI calculations are based on the avalanche year examples (box 2 of fig 2) but lines 241 et seq. indicate that RI values are also calculated for high magnitude events (Wit ≥ 0.3) only. Therefore, are there two sets of RI data for (i) avalanche years and (ii) high magnitude (Wit) years? So which data are used in the subsequent analyses? Are these high magnitude years simply binary data (yes/no?)
*Response*:
The return intervals are simply calculated for large magnitude avalanches, the only type of avalanche investigated in this study. There is only one set of RI values for each path, sub-region, and region. The avalanche years are binary. We added text (lines 244-245) to clarify that the RIs used throughout the study are the ones calculated after all processing steps.

*Comment*:
253 RAAIt is based on the definition of avalanche years (It), not high magnitude Wit years. Therefore avalanche years are identified using the It statistic but high magnitude avalanches are identified using the Wit statistic. It appears that the RI data are calculated based on both the It and the Wit classification whereas the RAAIt statistics are based on the It definition of avalanche years. Is this correct?
*Response*:
The RAAI is based on the $I_t$ index. The $W_{it}$ is simply a threshold to identify confidence in the signals. Once again, this illustrates the benefit of using high quality cross sections where most of the avalanche years we identified for each path using the thresholds developed by Corona et al. (2012) fell above the $W_{it}$ threshold.

*Comment*:
Line 324 seems to imply that the avalanche years identified in Figure 2 and the high magnitude events identified using Wit were identical so this difference does not matter? In any event only one set of calculations defining RI values should be specified. The term RI is used throughout the text but in places it is not clear whether it refers to the mean or median value.
*Response*:
We added text to clarify that we use median return interval throughout when referring to return interval (line 342).

*Comment*:
263 how does the probability of detection differ from the probability of avalanches?
*Response*:
The probability of detection (year) is a measure of the likelihood of detecting an avalanche year in the regional chronology by sampling any one given path and the probability of detection (path) is the probability of detecting the full chronology using any one given avalanche path. The probability of an avalanche would be the 1/RI (inverse of the return interval) for each individual path. These are both described in the revised Figure 2.

*Comment*:
284 are these comparisons included in this paper?
*Response*:
Yes. Line 348-349.

*Comment*:
291 ID by GD class but not type? So what was principal evidence used?
*Response*:
The GD class incorporates type in a systematic way for avalanche identification. Simply using type places imbalanced emphasis on certain types and not the cumulative signature of other types.

*Comment*:
298 this is predictable given the ages of trees sampled. Perhaps more interesting would be the years with the highest It values
*Response*:
The number of raw responses per year across all the paths is important as it provides a baseline to compare to avalanche years after applying signal:noise thresholds. The $I_t$ values simply provide a % of responses based on the number of trees alive in each year per individual path. This is simply used as a metric in the steps to identify avalanche years and don't serve to enhance the understanding of avalanche frequency when reported alone.

*Comment*:
299 Figures 3a and b appear identical and one is redundant. The scale on Fig 3b is incorrect (0.3 %?). The ages in Fig 3c indicate that many of these trees were quite small. What would the diameter of a 40-year old tree be? How does age influence the nature of the GD? In 3d were the larix and betula species identified?
*Response*:
We changed the figure to have one panel with the proportions labeled on the bars instead of a second panel. We didn't measure the diameter of each tree/sample as it wasn't relevant to the study. Instead, we chose samples that would have the capability of recording an avalanche event. Younger trees more pliable for a while so perhaps more resilient to impact pressure. Then as tree ages/grows it becomes more susceptible to uprooting by avalanches until it becomes larger/older. At that point the avalanche signal on an older tree is likely to be a scar or reaction wood. However, if the avalanche is sufficiently powerful, it will uproot the old/large tree. The Larix and Betula species were not identified given they were so few samples.

*Comment*:
308 missed 67 or 66?
*Response*:
Thanks for the catch. Changed to 67.

*Comment*:
312 Figure 4 needs a scale. Some comparison of the derived GD data would be useful to make the point. To be effective this topic warrants a more extensive discussion and presentation of data than that presented here. This discussion and figure should probably be deleted.
*Response*:
The scale in Figure 4 is labeled with the 5mm corer rectangles and mentioned in the caption. We believe that the results of this analysis in examining cores vs. cross-sections align with the objectives of the article. Determining the value of using cross-sections was central to the original sampling design which distinguishes this study from previous dendro-avalanche research. The exercise must be mentioned in the Methods section because we present values on the comparisons in the Results and then the Discussion. Lastly, we provide this comparison to quantitatively illustrate the difference in using cross sections vs. cores and do not in any way discount any studies that use cores.

*Comment*:
312 More importantly were results from these cores actually used in the analysis.

*Response*:
These "cores" were simulated cores as if we indeed cored the sample as opposed to using the full cross section.

*Comment*:
324 Table 5 what does the standard deviation figure refer to (bottom line). Explain in caption?
*Response*:
It refers to the std. dev. of the return interval. We added text to the caption to describe this.

*Comment*:
332 Tables 4 and 5 What is the statistic 1/RI in these tables? Why is the median RI value used rather than the mean? Explain in the caption.
*Response*:
1/RI refers to the probability of an avalanche occurring in that avalanche path in any given year. We added text to the caption describing this. We use the median as it is insensitive to outliers.

*Comment*:
334 Table 4 An additional line identifying the sub region should be added to the top of the table. The table should also give the period of record utilised to calculate RI for each track.
*Response*:
We added the line. We also added the period of record for each avalanche path. However, the POR for the return intervals was already listed and can be gleaned from avalanche years. The period of record (POR) for each path represents earliest inner year to the most recent outer year of all samples in the path. The RI was calculated on the return interval of avalanche years. This was added in the caption to current Table 3 (previous Table 4).

*Comment*:
337 JGO is a function of the early record but why LJB? LJB and LJC are 26, LGA is 25 and shed 7 is 28?
*Response*:
We don't really follow this comment. What are the values you reference for LJB, LJC, LGP (LGA [sic]) and S7? Those values aren't the RI for any of those paths.

*Comment*:
399 Table 7 explain MLC and HLC in the caption
*Response*:
Added text to updated Table 6.

*Comment*:
346 Figure 6 what are the data plotted in this Figure? The median of GDO in Table 4 is ca 34 but in this figure it is ca 28. For LGP the median is ca 12.6 in Table 4 but ca 8 in figure 6
*Response*:
Figure 6 shows the return intervals for each path, sub-region, and overall region as stated in the caption. Good catch. There is one typo in Table 4 (new Table 3). The median for JGO (GDO [sic]) is 28.5. However, the median for LGP is listed as 8 in Table 4 and is also 8 in Figure 6.

*Comment*:
338-40 surely the similarities and differences between tracks reflect the length and nature of the avalanche record in each track? Differences/ similarity in return intervals are partially dependant on the length of record
*Response*:

As we demonstrated to your comment in the beginning of the review, "scaling" the period of record makes a difference in only the two paths that we already discuss as being different in terms of RI values. Here is the response to that original comment: We subset the period of record for each path from 1967-2017 and compared RI values. Nine paths exhibit no change in RI values when compared to the full record and one path RI values decreased by 4 years. We observed larger changes in the other two paths; JGO path where only one avalanche year was recorded (down from 5) and the median RI in LJC changed from 22.5 years to 35 years. We previously discussed JGO and LJC and the variable RIs of each of those paths in the Discussion. This exercise highlights that discussion that these two paths were indeed slightly different than the others. We added the above text (ca. 348-351) to illustrate that we examined the most recent 50 years to "scale" the return periods to account for loss of older trees.

*Comment*:
349-66 These differences in recurrence intervals are calculated for different periods of record. To be comparable do they need to be calculated over the same interval?
*Response*:
See response above and, yes, we compared a similar period of record.

*Comment*:
369 The avalanche records in these tracks start in 1933, 1936 and 1993so why compare them to a record starting in 1908 which presumably has avalanches predating those records. Surely comparisons need to be over the same intervals?
*Response*:
See response above.

*Comment*:
370 Figure 7 a nice (original?) way to show these data.
*Response*:
Thanks.

*Comment*:
380 Figure 8b the red line is not visible. Perhaps delete it and simply indicate this value in the caption
*Response*:
We kept the line and the value in the caption. It is a bit difficult to see (hopefully the indicated value in the caption helps), but readers are able to zoom in a bit when viewing on a monitor and it's clearly evident then and provides a graphical reference for readers.

*Comment*:
386 et seq. But the records being compared have quite different lengths and histories. How unique is the record of individual paths? If you compared the record of the tracks with similar length of record (say, ca 1950-2010, RMA-C, 54.3, LJA, 10.7 and S4.7) how similar are they?
*Response*:
See responses above re: comparison of similar lengths of record.

*Comment*:
405-6 These trends are mainly an effect of the increased sampling of avalanche years
*Response*:
Yes, these trends are likely a function of increasing samples through time which is why we mention them, but don't hang our hat on the trend results. The RAAI is simply another way to view a regional chronology using techniques from previous literature to allow for comparison.

*Comment*:

432-3 relevance of these comparisons?
*Response*:
As per Reviewer 1's comments we removed these lines as they aren't necessarily relevant.

*Comment*:
435 up into the bottom? English? Is the bottom the end or center of the track?
*Response*:
Revised sentence to read "However, at several sites we also collected samples into the bottom of the track (S10.7, Shed 7, and 1163) rather than just the runout zone." (lines 447-448). The bottom is the end of the track just above the runout zone.

*Comment*:
437-40 some specific dates needed here as this is the basis for the selection of records used. What specifically is the most recent time period for which you have adequate data across the network?
*Response*:
This sentence is a bit confusing so we revised to read: "Therefore, we chose to examine more recent time periods dictated by the avalanche years identified through the double threshold methods." (lines 453-455).

*Comment*:
451-2 how frequent is tree removal? What % of GDs are termination of growth vs other indicators of avalanche damage?
*Response*:
We don't really know the frequency of tree removal. It depends on the impact pressure of any given avalanche and this isn't something we can tease out from our data. It is not possible to determine the real % of GDs due to termination of growth because we can't assume the tree was killed by an avalanche for all of our dead and downed samples. Tree mortality could be caused by insects, storm damage, etc. and a subsequent avalanche could then transport the tree. However, if we assume that all sampled trees were removed by an avalanche (a rather large assumption), then we can take the number of cross sections (614) divided by the number of GD (2134). This provides a rough estimate under this assumption.
$\frac{614}{2134} \times 100 = 29\%$ of GDs are termination of growth.

*Comment*:
454 Although mentioned several times this incomplete historical record is never presented or directly compared to the equivalent tree-ring record for the comparable sites
*Response*:
We state in lines 142-144 that we compare the records for the 3 paths in JFS Canyon to this historical observational record but only for qualitative purposes. The record is simply used to provide some context for 3 avalanche paths. We reference Reardon et al. (2008) and that is where the observational record can be found. Text from Methods: "We compared the reconstructed avalanche chronology of the JFS sub-region to the historical record for qualitative purposes of large magnitude years. A quantitative comparison would not be reflective of the true reliability of tree-ring methods because of the incomplete historical record."

*Comment*:
475 the difference between Readon's earlier results or the other avalanche tracks? What are these differences?
*Response*:
We revised the sentence for clarity to "This is likely the root of the difference for S10.7 and the reason this path contains the largest numbers of avalanche years in this analysis." (lines 487-488).

*Comment*:

481 LJC has the greatest RI? It has the greatest median but not mean. The large median is a function of the small sample size in this track. The fire may have taken out evidence for most events between 1943 and 2017 and therefore this is not a valid comparison.

*Response*:

We revised the sentence to read "...were the greatest in this sub-region.." (line 493) as the RIs for JGO are the greatest. We agree that the fire played a major role in removing some evidence and now the slope is more exposed and susceptible to avalanching. We added text reflecting the fire's impact on data availability (line 500).

*Comment*:

489 using which RI value, mean or median? What is the correlation statistic?

*Response*:

We compared both mean and median RI values. $r$=0.65, p=0.02, Figure A1. This was stated in the Results (line 348-349 previously lines 343-344).

*Comment*:

495 JGO is very unusual with only two avalanches between 1880 and 2017! The critical difference is the absence of documented events in the last 50 years. The only answer to these tentative explanations is more data from adjacent tracks. Perhaps the only comment necessary is that the reason for this is not known.

*Response*:

This path is unusual and we provide some explanation given its unique geographical location east of the Continental Divide within our dataset. We added text to reflect your very good points about this path (lines 510-511). "To understand if this value is accurate, we would have to sample adjacent tracks to determine if the return intervals are similar or not."

*Comment*:

507 but these differences are never explored

*Response*:

As previously mentioned, it is beyond the scope of this paper to explore the localized weather and climate drivers and the interaction with terrain. We explore such atmospheric and climate drivers in another manuscript.

*Comment*:

527 but these changes are also influenced by which avalanche track you remove.

*Response*:

The changes are only influenced by the removal (or addition) of the S10.7 path which we state and discuss in the next paragraph. We also reference recent literature that discusses the importance of selecting individual avalanche paths (line 550-552).

*Comment*:

532 But how typical is the record of s10-7 of other paths in the region see e.g. Table 4

*Response*:

We added text describing how S10.7 differs in line 552-553. "This is also illustrated by the large number of avalanche years detected in S10.7 due to increased sampling in the track."

*Comment*:

533 s10.7 has the most avalanche activity but surprisingly is not compared with the available, if limited, observational record.

*Response*:

Reardon et al. (2008) provides a more detailed examination of the individual path S10.7 and by referencing their work we are able to focus on the effect to the overall regional chronology, the major objective of our study.

*Comment*:
538 What is an avalanche cycle chronology?
*Response*:
Including the word cycle shows that, at the regional scale, we are able to capture major avalanche cycles (widespread avalanche event) through time. We added "(widespread avalanche event)" to line 558.

*Comment*:
547-9 these data would be useful here to validate some of these comments or are they solely based on the examples which follow.
*Response*:
They are based on the examples that follow.

*Comment*:
566 paths with one scar or one GD of class 3? Where are these scar data?
*Response*:
Good catch. We changed to "GD" (line 586). We referenced the data in Section 7 - Data Availability (lines 651-655) (https://doi.org/10.5066/P9TLHZAI, 2019)

*Comment*:
570 is the sample design or the number of paths the critical factor here? The sample design clearly increases the area covered.
*Response*:
We discuss that sampling more paths certainly increases the POD of avalanche years. However, the sampling design using scale triplet allows one to scale the process of avalanching from small path scale to the larger regional scale.

*Comment*:
582-5 Basic point is that if you sample more avalanche tracks you get more avalanche years and a more consistent pattern may emerge. However, the pattern of avalanche activity varies from track to track and from year to year.
*Response*:
Correct. This illustrates the benefit of such a sampling design where one can scale the process across spatial extents.

*Comment*:
587 Is this a function of sample size or other characteristics such as the time period covered by those samples and the sampling network?
*Response*:
A function of sample size. We collected a large number of samples across the region, but at the individual path scale, more would have been better in two of the paths.

*Comment*:
603 this median value probably should be linked to a time frame to which it applies
*Response*:
We added the full regional chronology period of record (1866-2017) (line 623).

*Comment*:

Figure A1 What are the data used here (reference to Table 2)? Some numbers are barely visible. Perhaps use bolder (larger) numbers and colour as a background to individual cells?
*Response*:
Yes, the data refer to Table 2 (new Table 1). We revised Figure A1 with larger numbers and a different background for easier readability.

---

## Author Comment (AC3) · 24 Oct 2020

Thanks for this note and no problem!
* * *

---

## Referee Report (RR1)

Brian Luckman (referee)
Luckman@uwo.ca
November 23rd, 2020

**Overall comment**

I reiterate my general comments from the initial review. *This paper presents snow avalanche histories from 12 avalanche tracks, 3 from each of four regions in the Northern Rocky Mountains of N.W Montana based on tree-ring data from 637 trees. These data are used to define the history and frequency of large magnitude avalanches for individual tracks, sub-regions (mountain ranges) and across a region of ca 3500 km². The paper then estimates the efficiency of using various combinations of these chronologies to estimate a regional chronology of high magnitude avalanches in order to guide future sampling strategies for estimating regional avalanche activity. The techniques used are based on prior usage from the literature and, to my knowledge, the attempt to assess the efficiency of developing a regional history is novel*

The authors have addressed many of the issues raised in my earlier review. However, I still maintain that there needs to be some indication of the relative importance of the various inputs (GDs) to the determination of growth disturbances.
 Secondly there needs to be a stronger recognition that although significant large avalanches can be identified  from the longer tree ring records, any assessment or comparison of recurrence intervals must be based on an adequate sample base i.e. one without significant gaps due to the influence of fire or probable removal of parts of the record by previous avalanche activity. Therefore there should be some assessment of the quality of the record from individual tracks and the discussion of recurrence intervals should be restricted to their "scaled" data set.

The authors provided a set of responses to my initial review comments plus a revised manuscript (identified herein as ms2). Attached please find two sets of comments
(i) my comments on several of these responses prior to evaluation of the revised ms (*text in italics is my responses)*
(ii) Comments on the revised manuscript.

**PART 1 responses to previous comments**

**BHL Comment:** One of the principal difficulties is the comparison of statistics such as RI values between sites based on records of different length where the RI values are strongly related to survival of older individuals within the avalanche path. Perhaps a

comparison based on e.g. the last fifty years would be better to compare differences between tracks. **AU Response:** We subset the period of record for each path from 1967-2017 and compared RI values. *See Comments on ms2*

**BHL Comment.** There is no specific exploration of the relationship between avalanche activity and climatic factors.

**AU Response:** Climate and regional avalanche relationships are the topic of a follow-on manuscript using this dataset that is currently undergoing peer-review.

*In this regard, it would be interesting to know what proportion of the avalanches are "direct action" i.e. directly triggered by precipitation vs delayed action due to changes in the condition of the snowpack since the former are more directly controlled by ( regional) precipitation events*

**Detailed and specific comments:**

**BHL Comment:** 312 More importantly were results from these cores actually used in the analysis.

**AU Response:** These "cores" were simulated cores as if we indeed cored the sample as opposed to using the full cross section. *This is a misunderstanding. The question was rather were data from the cored trees used in the analysis (not the simulated cores from the cross section examples).*

**BHL Comment:** 334 Table 4 An additional line identifying the sub region should be added to the top of the table.

**AU Response:** We added the line. *This refers to the new Table 3 and this line was not added*

**BHL Comment:** 337 JGO is a function of the early record but why LJB? LJB and LJC are 26, LGA is 25 and shed 7 is 28?

**AU Response:** We don't really follow this comment. What are the values you reference for LJB, LJC, LGP (LGA [sic]) and S7? Those values aren't the RI for any of those paths.

*Values are the difference between the max and min values in old Table 4 JGO is 68*

**BHL Comment:** 338-40 surely the similarities and differences between tracks reflect the length and nature of the avalanche record in each track? Differences/ similarity in return intervals are partially dependant on the length of record

**AU Response:** As we demonstrated to your comment in the beginning of the review, "scaling" the period of record makes a difference in only the two paths that we already discuss as being different in terms of RI values. Here is the response to that original comment: We subset the period of record for each path from 1967- 2017 and compared RI values. Nine paths exhibit no change in RI values when compared to the full record and one path RI values decreased by 4 years. We observed larger changes in the other two paths; JGO path where only one avalanche year was recorded (down from 5) and the median RI in LJC changed from 22.5 years to 35 years. We previously discussed JGO and LJC and the variable RIs of each of those paths in the Discussion. This exercise highlights that discussion that these two paths were indeed slightly different than the others. We added the above text (ca. 348-351) to illustrate that we examined the most recent 50 years to "scale" the return periods to account for loss of older trees.

*The lack of change between the original and "scaled data" is largely because of the similarity of records between the two data sets as most sites only have records for the period after 1967. It does not address the gaps in several records.*

**BHL Comment:** 405-6 These trends are mainly an effect of the increased sampling of avalanche years

**AU Response:** Yes, these trends are likely a function of increasing samples through time which is why we mention them, but don't hang our hat on the trend results. The RAAI is simply another way to view a regional chronology using techniques from previous literature to allow for comparison.

*The point here is that the trend has nothing to do with trends in avalanche activity but is mainly due to the increased availability of sites and sampling over time and should therefore be deleted (lines 420-424 in the 2ⁿᵈ ms)*

**BHL Comment:** 435 up into the bottom? English? Is the bottom the end or center of the track?

**AU Response:** Revised sentence to read "However, at several sites we also collected samples into the bottom of the track (S10.7, Shed 7, and 1163) rather than just the runout zone." (lines 447-448). The bottom is the end of the track just above the runout zone.

*451-2 (ms2)     Still not clear to me. How is the end (bottom) of the track different from the runout zone? From the context it would appear that the bottom here refers to the middle of the track upvalley (i.e. having smaller avalanches) of the runout zone*

**BHL Comment:** 451-2 how frequent is tree removal? What % of GDs are termination of growth vs other indicators of avalanche damage?

**AU Response:** We don't really know the frequency of tree removal. It depends on the impact pressure of any given avalanche and this isn't something we can tease out from our data. It is not possible to determine the real % of GDs due to termination of growth because we can't assume the tree was killed by an avalanche for all of our dead and downed samples. *This could be tested by comparison of outer ring dates with known dates of avalanches (based on other evidence) in the track.* Tree mortality could be caused by insects, storm damage, etc. and a subsequent avalanche could then transport the tree. However, if we assume that all sampled trees were removed by an avalanche (a rather large assumption), then we can take the number of cross sections (614) divided by the number of GD (2134). This provides a rough estimate under this assumption. $614/2134 \times 100 = 29\%$ of GDs are termination of growth. *This is not clear. You clearly identified some outer dates as termination of growth by avalanches and not others. So what criteria were used if you scored termination of growth as a GD.?*

**BHL Comment:** 538 What is an avalanche cycle chronology?

**AU Response:** Including the word cycle shows that, at the regional scale, we are able to capture major avalanche cycles (widespread avalanche event) through time. We added "(widespread avalanche event)" to line 558. *Cycles is the wrong word*

**PART 2 Comments on the revised paper (line numbers as per ms2)**

**Line    comment**
**151**    within the previous
**156**    also depends on the available length of record in the trees sampled.

**185**   verified dating against?

**205**      Classification of GDs. The response to my earlier comment on the need for information about the number and types of GDs was as follows "The GD class incorporates type in a systematic way for avalanche identification. Simply using type places imbalanced emphasis on certain types and not the cumulative signature of other types". However, although GDs as defined do summarise the quality of evidence in an individual year I maintain that it is important for the reader to understand what proportion of the GDs in classes 4 and 5 were based primarily on scars vis-a-vis other criteria. The selection of sites for cross sections suggests that scars were of primary importance in obtaining evidence of past avalanche events.

**236**      scars or injuries = GDs? The text still does not indicate the approximate nature of these GD values i.e. how many of the GD>3 were based on primarily on scars, reaction wood, TRDs, etc or alternatively the combination of several lines of evidence for the same year. This is important for readers to ascertain the relative significance of the principal lines of evidence on which these chronologies are based.

**242-3**  "We use these RI derived after filtering events for confidence as the intervals throughout the study" .clarify? = "We use these RI values determined after filtering events throughout the study"

**274-9**   the results of this analysis are discussed later (line 420, see below) where my comment is that they have little or no value for discussion of changes in avalanche activity. Therefore this text should be removed.

**420-4**   The point here is that the trend has nothing to do with trends in avalanche activity but is mainly due to the increased availability of sites and sampling over time and should therefore be deleted.

**298**      The accepted abbreviation for *P. engelmannii* is PCEN not PIEN (PI=pinus). See also Fig 3 caption and label lowest axis Fig 3

**306**      Betul =betula?

Section 3.1. I remain to be convinced of this analysis. You should at least provide both sets of summary data.

**323-4**   invert order of sentence "...no clear pattern of similarly identified years from paths….. "

**326**      weighted?

Table 3  (old Table 4) a line identifying the sub regions was not added.

This table clearly indicates the strong differences in the reconstructed avalanche histories from these tracks. The only sites that would appear to have a relatively comparable record are the three Red Meadow sites. The others rarely overlap or have unique characteristics (e.g. shed 10-7).

Track LGP has only 3 avalanches.  In the table max and min RI are 30 and 8 not 27 and 3 --  and how can one have a median of 8 and mean of 12.67 from two data points?

**343**      these sub regions not identified in Table 3

**345**      GTSR is the most similar sub region? But in table 3 WF region has medians of 3, 5 and 8, GTSR has medians of 8, 14 and 28.5.

**353**       probably because these paths have a reasonable record over this interval whereas 54-3, JGO and LJC are demonstratively different.

**352-7** The similarity between results from the "truncated" and complete records is because the records in the "truncated" and complete records are basically similar when the unusual sites are deleted.

Table 6 column 3; 24+1=27? Column 6;22+11=34? (included+ excluded = total)

**403** LGP has the next greatest sample size? Trees sampled or events? LGP only has 3 avalanche events in Table 3?

**427** so scars are identified as such?

**436** not surprising as these are the most similar and consistent records.

**450** collection from areas?

**468-9** 10-50% is a large range. Some data should be provided to support this comment or it should be deleted.

**490** It is not clear whether the difference is between (i) the Reardon results (not given) and those for path S10.7 in this paper or (ii) between path S10.7 and other tracks in this paper.

**499** The problem here is that these large RI values reflect the irregular preservation of evidence for large avalanches. One is sampling a truncated distribution with gaps in the evidence due to removal by intervening avalanche or fire events and, in a single track, these cannot be differentiated from gaps in avalanche activity. Therefore it is not possible to distinguish whether these large RI values are real or an artefact of the preservation of data. Consequently the subsequent discussion of possible causes for the lack of large magnitude avalanches in the JGO track are invalid because of the limited sampling of sites east of the divide .

The real problem seems to be that, in order to provide relatively secure estimates of recurrence intervals of large avalanches one needs an appropriate sample base without obvious temporal gaps. As one goes back in time this becomes increasingly difficult. Therefore there needs to be explicit evaluation of the records in some of these sites and the results from sites with limited sample depth should be treated with caution.

**515** quantitative data to support this?

**525-6** and also local avalanche/ stand/ fire history.

**530-1** it is also important to establish the relative importance of avalanche triggers (i.e. direct vs delayed action avalanches) when establishing relationships with climatic controls.

**586** use of cycle (see earlier comment on ms1) replace cycle with year

**602** OK see earlier comment on line 420

**Table A3** Modify 2nd column heading and reduce width of table to two columns -

**Figure A1** Statistical significance of the values? Caption should indicate source data are in Table 1.

---

## Author Response (AR2)

**Authors' response to 2nd set of comments by Brian Luckman**

We thank Brian Luckman for reviewing the revised manuscript and providing comments to help improve this manuscript. We are pleased that the reviewer recommends publication after minor revisions. We address the second set of comments on the revised version (ms2) below.

*Overall comment*
I reiterate my general comments from the initial review. *This paper presents snow avalanche histories from 12 avalanche tracks, 3 from each of four regions in the Northern Rocky Mountains of N.W Montana based on tree-ring data from 637 trees. These data are used to define the history and frequency of large magnitude avalanches for individual tracks, sub-regions (mountain ranges) and across a region of ca 3500 km². The paper then estimates the efficiency of using various combinations of these chronologies to estimate a regional chronology of high magnitude avalanches in order to guide future sampling strategies for estimating regional avalanche activity. The techniques used are based on prior usage from the literature and, to my knowledge, the attempt to assess the efficiency of developing a regional history is novel*

The authors have addressed many of the issues raised in my earlier review. However, I still maintain that there needs to be some indication of the relative importance of the various inputs (GDs) to the determination of growth disturbances.
*Peitzsch et al. Response:*
We included a table (Table A3) in Appendix A indicating the proportion of each growth disturbance signal to each growth disturbance class used in this study. This illustrates the relative importance of each signal to the final GD class. We also included a sentence in the Results (line 288) and elaborated on this in the Discussion (lines 476).

Secondly there needs to be a stronger recognition that although significant large avalanches can be identified from the longer tree ring records, any assessment or comparison of recurrence intervals must be based on an adequate sample base i.e. one without significant gaps due to the influence of fire or probable removal of parts of the record by previous avalanche activity. Therefore there should be some assessment of the quality of the record from individual tracks and the discussion of recurrence intervals should be restricted to their "scaled" data set.
*Peitzsch et al. Response:*
We included further discussion of the "scaled" dataset and included an assessment of the quality of records from individual paths (line 491). Revised to: "The results from examining return intervals during a truncated period from 1967-2017 across all paths illustrate that several of the individual path return interval results should be treated with caution (e.g. JGO, LJC, and LGP). The difference in minimum and maximum return interval values is a function of a decreasing sample size back in time. The minimum return interval values in many of the paths are concentrated during recent periods. This is a limitation of using dendrochronology to estimate return intervals. Comparing avalanche return intervals across individual paths should also be treated with caution given the variable nature of sample availability across paths. This variability across individual paths further provides reason to evaluate the number of paths necessary to create a regional avalanche chronology from tree rings. Most of the paths have a reasonable record over this truncated period and also highlight the importance of strategic sampling in numerous avalanche paths. While dendrochronology underestimates avalanche activity, we show that sampling enough paths across a region provides a reasonable estimate of avalanche activity at this scale."

The authors provided a set of responses to my initial review comments plus a revised manuscript (identified herein as ms2). Attached please find two sets of comments

*(i)* my comments on several of these responses prior to evaluation of the revised ms
(*text in italics is my responses*)
(ii) Comments on the revised manuscript.

**PART 1 responses to previous comments**

**BHL Comment:** One of the principal difficulties is the comparison of statistics such as RI values between sites based on records of different length where the RI values are strongly related to survival of older individuals within the avalanche path. Perhaps a comparison based on e.g. the last fifty years would be better to compare differences between tracks. **AU Response:** We subset the period of record for each path from 1967- 2017 and compared RI values. *See Comments on ms2*
*Peitzsch et al. Response:*

Addressed in comments on ms2.

**BHL Comment**. There is no specific exploration of the relationship between avalanche activity and climatic factors.
**AU Response:** Climate and regional avalanche relationships are the topic of a follow-on manuscript using this dataset that is currently undergoing peer-review.
*In this regard, it would be interesting to know what proportion of the avalanches are "direct action" i.e. directly triggered by precipitation vs delayed action due to changes in the condition of the snowpack since the former are more directly controlled by ( regional) precipitation events*
*Peitzsch et al. Response:*

Noted. Thanks for the suggestion.

**Detailed and specific comments**:
**BHL Comment:** 312 More importantly were results from these cores actually used in the analysis.
**AU Response:** These "cores" were simulated cores as if we indeed cored the sample as opposed to using the full cross section. *This is a misunderstanding. The question was rather were data from the cored trees used in the analysis (not the simulated cores from the cross section examples).*
*Peitzsch et al. Response:*

Cores were used in the overall analysis. See line 287.

**BHL Comment:** 334 Table 4 An additional line identifying the sub region should be added to the top of the table.
**AU Response:** We added the line. *This refers to the new Table 3 and this line was not added*
*Peitzsch et al. Response:*

We now added the sub-region at the top of Table 3.

**BHL Comment:** 337 JGO is a function of the early record but why LJB? LJB and LJC are 26, LGA is 25 and shed 7 is 28?
**AU Response:** We don't really follow this comment. What are the values you reference for LJB, LJC, LGP (LGA [sic]) and S7? Those values aren't the RI for any of those paths.
*Values are the difference between the max and min values in old Table 4 JGO is 68*

*Peitzsch et al. Response:*

Thanks for clarifying. We addressed this issue with more explanation in the discussion of the "scaled" RIs for the individual paths (line 491)

**BHL Comment:** 338-40 surely the similarities and differences between tracks reflect the length and nature of the avalanche record in each track? Differences/ similarity in return intervals are partially dependant on the length of record.

**AU Response:** As we demonstrated to your comment in the beginning of the review, "scaling" the period of record makes a difference in only the two paths that we already discuss as being different in terms of RI values. Here is the response to that original comment: We subset the period of record for each path from 1967- 2017 and compared RI values. Nine paths exhibit no change in RI values when compared to the full record and one path RI values decreased by 4 years. We observed larger changes in the other two paths; JGO path where only one avalanche year was recorded (down from 5) and the median RI in LJC changed from 22.5 years to 35 years. We previously discussed JGO and LJC and the variable RIs of each of those paths in the Discussion. This exercise highlights that discussion that these two paths were indeed slightly different than the others. We added the above text (ca. 348-351) to illustrate that we examined the most recent 50 years to "scale" the return periods to account for loss of older trees.

*The lack of change between the original and "scaled data" is largely because of the similarity of records between the two data sets as most sites only have records for the period after 1967. It does not address the gaps in several records.*

*Peitzsch et al. Response:*

We chose to scale the data to this time period precisely because of this. This allows a comparison with the greatest number of responses. The gaps are indeed due to other factors which we now address in the discussion in the updated version of the manuscript. See responses below and above (and line numbers) to specific comments in the new manuscript on this topic of RIs.

**BHL Comment:** 405-6 These trends are mainly an effect of the increased sampling of avalanche years

**AU Response:** Yes, these trends are likely a function of increasing samples through time which is why we mention them, but don't hang our hat on the trend results. The RAAI is simply another way to view a regional chronology using techniques from previous literature to allow for comparison.

*The point here is that the trend has nothing to do with trends in avalanche activity but is mainly due to the increased availability of sites and sampling over time and should therefore be deleted (lines 420-424 in the 2$^{nd}$ ms)*

*Peitzsch et al. Response:*

We deleted the reference to the trends in the Results as well as the Methods and Discussion.

**BHL Comment:** 435 up into the bottom? English? Is the bottom the end or center of the track?

AU **Response:** Revised sentence to read "However, at several sites we also collected samples into the bottom of the track (S10.7, Shed 7, and 1163) rather than just the runout zone." (lines 447-448). The bottom is the end of the track just above the runout zone.

*451-2 (ms2)    Still not clear to me. How is the end (bottom) of the track different from the runout zone? From the context it would appear that the bottom here refers to the middle of the track upvalley (i.e. having smaller avalanches) of the runout zone*

*Peitzsch et al. Response:*

We revised this sentence for clarity:
"At several sites we collected samples at the upper extent of the runout zones (S10.7, Shed 7, and 1163)." (line 444)

**BHL Comment:** 451-2 how frequent is tree removal? What % of GDs are termination of growth vs other indicators of avalanche damage?

**AU Response:** We don't really know the frequency of tree removal. It depends on the impact pressure of any given avalanche and this isn't something we can tease out from our data. It is not possible to determine the real % of GDs due to termination of growth because we can't assume the tree was killed by an avalanche for all of our dead and downed samples. *This could be tested by comparison of outer ring dates with known dates of avalanches (based on other evidence) in the track.* Tree mortality could be caused by insects, storm damage, etc. and a subsequent avalanche could then transport the tree. However, if we assume that all sampled trees were removed by an avalanche (a rather large assumption), then we can take the number of cross sections (614) divided by the number of GD (2134). This provides a rough estimate under this assumption. $614/2134 \times 100 = 29\%$ of GDs are termination of growth. *This is not clear. You clearly identified some outer dates as termination of growth by avalanches and not others. So what criteria were used if you scored termination of growth as a GD.?*

Peitzsch et al. Response:

We included termination of growth due to avalanches based on the historical/observational record as is classified as a $C_1$ response. This is also included in the new Table A3 as "Termination of Growth". Some of the samples even have earlywood when the avalanche occurred in the late winter or early spring. However, when we have termination of growth that does not coincide with a known avalanche year from the observational record we cannot assume it was killed by an avalanche. So the proportion included in Table A3 is when termination of growth coincides with a known (observed) avalanche.

**BHL Comment:** 538 What is an avalanche cycle chronology?

**AU Response:** Including the word cycle shows that, at the regional scale, we are able to capture major avalanche cycles (widespread avalanche event) through time. We added "(widespread avalanche event)" to line 558. *Cycles is the wrong word*

Peitzsch et al. Response:

We removed the word cycle and replaced with "year". See comment below in new ms.

**PART 2 Comments on the revised paper (line numbers as per**

**ms2) Line    comment**
**151**    within the previous
Response:
Revised. (line 150)

**156**    also depends on the available length of record in the trees sampled.
Response:
Added " This also depends on the available length of record within a given avalanche path." (line 156)

**185**    verified dating against?
Response:
Added "dating" to read: "We assessed cross-dating calendar-year accuracy of each sample

and statistically verified dating against measured samples..." (line 185)

**205** Classification of GDs. The response to my earlier comment on the need for information about the number and types of GDs was as follows "The GD class incorporates type in a systematic way for avalanche identification. Simply using type places imbalanced emphasis on certain types and not the cumulative signature of other types". However, although GDs as defined do summarise the quality of evidence in an individual year I maintain that it is important for the reader to understand what proportion of the GDs in classes 4 and 5 were based primarily on scars vis-a-vis other criteria. The selection of sites for cross sections suggests that scars were of primary importance in obtaining evidence of past avalanche events.
*Response:*
See response above where we include this table in Appendix A, the Results (line 288) and elaborate in the Discussion. (lines 476)

**236** scars or injuries = GDs? The text still does not indicate the approximate nature of these GD values i.e. how many of the GD>3 were based on primarily on scars, reaction wood, TRDs, etc or alternatively the combination of several lines of evidence for the same year. This is important for readers to ascertain the relative significance of the principal lines of evidence on which these chronologies are based.
*Response:*
See response above where we include this table in Appendix A, the Results (line 288) and elaborate in the Discussion. (lines 476)

**242-3** "We use these RI derived after filtering events for confidence as the intervals throughout the study" .clarify? = "We use these RI values determined after filtering events throughout the study"
*Response:*
Revised to: "We use these RI values determined after filtering events throughout the study." (line 242)

**274-9** the results of this analysis are discussed later (line 420, see below) where my comment is that they have little or no value for discussion of changes in avalanche activity. Therefore this text should be removed.
**420-4** The point here is that the trend has nothing to do with trends in avalanche activity but is mainly due to the increased availability of sites and sampling over time and should therefore be deleted.
*Response:*
To both comments: see response above. We removed this text throughout the manuscript.

**298** The accepted abbreviation for *P. engelmannii* is PCEN not PIEN (PI=pinus). See also Fig 3 caption and label lowest axis Fig 3
*Response:*
Thanks for catching that. We revised both to PCEN.

**306** Betul =betula?
*Response:*
Thanks for catching that as well. We revised to BETULA in axis and caption.

Section 3.1. I remain to be convinced of this analysis. You should at least provide both sets of summary data.

*Response:*

We included summary data in new Table A4. We also want to emphasize that we are not suggesting that one must use only cross sections. We added this to the Discussion "We do not discount any studies that use cores for reconstructing avalanche chronologies and understand there are sampling limitations from environmental and policy perspectives in different regions as well as financial and processing constraints. However, we are suggesting that if the ability to collect cross sections exists, then it is advantageous to collect them."

(line 439)

**323-4**   invert order of sentence "...no clear pattern of similarly identified years from paths….. "

*Response:*

Revised to "There was no clear pattern of similarly identified years from paths physically closer in proximity to each other." (line 317)

**326**   weighted?

*Response:*

Revised to "When we applied the $W_{it}$ process step, the number of identified avalanche years did not change for any individual avalanche path compared to application of the double threshold method alone." (line 319)

Table 3  (old Table 4) a line identifying the sub regions was not added.
This table clearly indicates the strong differences in the reconstructed avalanche histories from these tracks. The only sites that would appear to have a relatively comparable record are the three Red Meadow sites. The others rarely overlap or have unique characteristics (e.g. shed 10-7).

*Response:*

Added sub-regions heading to Table 3.
We revised the discussion to reflect this (line 491). We also maintain the comparisons as evidence to the overall objective in the study, which is to evaluate the notion of "regional" avalanche chronology from tree rings.

Track LGP has only 3 avalanches. In the table max and min RI are 30 and 8 not 27 and 3 -- and how can one have a median of 8 and mean of 12.67 from two data points?

*Response:*

When revising this table we mistakenly left out 1974 in LGP. We added that to Table 3.

**343**   these sub regions not identified in Table 3

*Response:*

Added sub-regions heading to Table 3.

**345**   GTSR is the most similar sub region? But in table 3 WF region has medians of 3, 5 and 8, GTSR has medians of 8, 14 and 28.5.

*Response:*

We revised to read "The avalanche paths within the WF sub-region had the most similar return intervals of any of the sub-regions." (line 339)

**353**   probably because these paths have a reasonable record over this interval

whereas 54-3, JGO and LJC are demonstratively different.
*Response:*
Added text to the Discussion (line 491).

**352-7** The similarity between results from the "truncated" and complete records is because the records in the "truncated" and complete records are basically similar when the unusual sites are deleted.
*Response:*
Added "If we removed 54-3, JGO and LJC for this comparison, the records from the subset period of record are similar to the complete records for the other paths in the study." (line 349)

**Table 6** column 3; 24+1=27? Column 6;22+11=34? (included+ excluded = total)
*Response:*
To clarify we added this to Table 6 caption:
"'# not in regional' refers to avalanche years identified in that particular combination of paths but not identified in the regional record."

**403** LGP has the next greatest sample size? Trees sampled or events? LGP only has 3 avalanche events in Table 3?
*Response:*
Sample size.
"...(the other path with the greatest size of sampled trees)..." (line 397)

**427**    so scars are identified as such?
*Response:*
Changed to "When we examined avalanche paths that exhibited at least one GD during avalanche years identified in the regional chronologies (i.e. no thresholds used), the *POD* is generally greater." (line 415)

**436**    not surprising as these are the most similar and consistent records.
*Response:*
Added this to the Discussion: "The WF sub-region captured the regional chronology most consistently because of the similar and consistent records within the sub-region." (line 596)

**450**    collection from areas?
*Response:*
Revised to "We targeted sample collection in the runout zones..." (line 443)

**468-9** 10-50% is a large range. Some data should be provided to support this comment or it should be deleted.
*Response:*
We deleted this as the historical record is incomplete which is the reason the range is large.

**490**    It is not clear whether the difference is between (i) the Reardon results (not given) and those for path S10.7 in this paper or (ii) between path S10.7 and other tracks in this paper.
*Response:*

Revised to "This is likely the root of the difference for S10.7 and the other paths in this study and..." (line 485)

**499** The problem here is that these large RI values reflect the irregular preservation of evidence for large avalanches. One is sampling a truncated distribution with gaps in the evidence due to removal by intervening avalanche or fire events and, in a single track, these cannot be differentiated from gaps in avalanche activity. Therefore it is not possible to distinguish whether these large RI values are real or an artefact of the preservation of data. Consequently the subsequent discussion of possible causes for the lack of large magnitude avalanches in the JGO track are invalid because of the limited sampling of sites east of the divide.
The real problem seems to be that, in order to provide relatively secure estimates of recurrence intervals of large avalanches one needs an appropriate sample base without obvious temporal gaps. As one goes back in time this becomes increasingly difficult. Therefore there needs to be explicit evaluation of the records in some of these sites and the results from sites with limited sample depth should be treated with caution.
*Response:*
Removed discussion of potential climatic reasons for JGO and revised to:
"JGO contains the maximum return interval for any path in the study, and the return intervals are significantly different than numerous other paths. A lack of recording data after one large avalanche event could easily skew this value. To understand if this value is accurate, we would have to sample adjacent tracks to determine if the return intervals are similar or not. An appropriate sample base without large temporal gaps is necessary to fully provide an accurate estimate of return intervals within a single avalanche path. While the sample size is sufficient for this individual path, the results should be treated with caution due to the temporal gaps. In other words, the large return interval values may reflect the irregular preservation of evidence for large avalanches as opposed to an accurate estimate of return intervals. Therefore, we cannot fully explain the large maximum return interval for this path." (line 502)

**515** quantitative data to support this?
*Response:*
See line 343 and Figure A1.

**525-6** and also local avalanche/ stand/ fire history.
*Response:*
"The differences between individual avalanche paths as well as sub-regions are likely due to localized terrain and weather/climate factors and the interaction of the two (Chesley-Preston, 2010) as well as local avalanche, forest stand, and fire history." (line 530)

**530-1** it is also important to establish the relative importance of avalanche triggers (i.e. direct vs delayed action avalanches) when establishing relationships with climatic controls.
*Response:*
Yes. The relationship between direct action avalanches and spatial extent across a region vs. delayed action avalanches within avalanche paths across a region in the context of climatic controls is an important question for future work.

**586** use of cycle (see earlier comment on ms1) replace cycle with year
*Response:*
Changed to "year" (line 593).

**602** OK see earlier comment on line 420

*Response:*
See response above. Removed this from manuscript.

**Table A3** Modify 2nd column heading and reduce width of table to two columns –
*Response:*
Modified 2nd column and centered text which is the reason it looked like 3 columns.

**Figure A1** Statistical significance of the values? Caption should indicate source data are in Table 1.
*Response:*
Added to caption of Figure A1: "Statistical significance is $p < 0.05$. See source data in Table 1."